# The Genus *Paronychia* (Caryophyllaceae) in South America: Nomenclatural Review and Taxonomic Notes with the Description of a New Species from North Peru

**DOI:** 10.3390/plants12051064

**Published:** 2023-02-27

**Authors:** Duilio Iamonico, Daniel B. Montesinos-Tubée

**Affiliations:** 1Department of Environmental Biology, University of Rome Sapienza, Piazzale Alod Moro 5, 00185 Rome, Italy; 2Instituto Científico Michael Owen Dillon, Av. Jorge Chávez 610, Cercado, Arequipa 04001, Peru; 3Instituto de Ciencia y Gestión Ambiental de la Universidad Nacional de San Agustín de Arequipa, Calle San Agustín 108, Arequipa 04001, Peru; 4Naturalis Biodiversity Centre, Darwinweg 2, 2333 CR Leiden, The Netherlands

**Keywords:** America, lectotype, neotype, new species, nomenclatural change, typification

## Abstract

All the names in *Paronychia* described from South America are investigated. Five names (*P. arbuscula*, *P. brasiliana* subsp. *brasiliana* var. *pubescens*, *P. coquimbensis*, *P. hieronymi*, and *P. mandoniana*) are lecto- or neotypified on specimens preserved at GOET, K, LP, and P. The typification of nine names, first proposed by Chaudhri in 1968 as the “holotype” are corrected according to Art. 9.10 of ICN. Three second-step typifications (Art. 9.17 of ICN) are proposed for *P. camphorosmoides*, *P. communis*, and *P. hartwegiana*. The following nomenclatural changes are proposed: *P. arequipensis comb. et stat. nov.* (basionym: *P. microphylla* subsp. *microphylla* var. *arequepensis*), *P. compacta nom. nov. pro P. andina* (Philippi non Gray; Art. 53.1 of ICN), *P. jujuyensis comb. et stat. nov.* (basionym: *P. hieronymi* subsp. *hieronymi* var. *jujuyensis*), *P. compacta* subsp. *boliviana comb. nov.* (basionym: *P. andina* subsp. *boliviana*), and *P. compacta* subsp. *purpurea comb. nov.* (basionym: *P. andina* subsp. *purpurea*). A new species (*P. glabra sp. nov.*) is proposed based on our examination of live plants and herbarium specimens. *P. johnstonii* subsp. *johnstonii* var. *scabrida* is synonymized (*syn. nov.*) with *P. johnstonii*. Finally, *P. argyrocoma* subsp. *argyrocoma* is excluded from South America since it was based on misidentified specimens (deposited at MO) of *P. andina* subsp. *andina*. A total of 30 species (43 taxa including subspecies, varieties, subvarieties, and forms) are recognized, highlighting that for some (*Paronychia chilensis*, *P. communis*, *P. setigera*) we provisionally accept Chaudhri’s infraspecific classification, since the high phenotypic variability of these taxa is quite complicated and further investigations need to solve their taxonomy.

## 1. Introduction

*Paronychia* Mill. (Paronychieae Dumort., Paronychioideae Meisn., Caryophyllaceae Juss.) is a genus of 110–120 species native to temperate, tropical, and subtropical regions of America and Africa, Macaronesia, the Mediterranean area, and the Middle East [1,2]. On the basis of molecular analyses [3], subfam. Paronychioideae is paraphyletic [4] with the genera *Spergula* Dill. ex L. and *Spergularia* (Pers.) J.Presl & C.Presl to be excluded (see also [5]); these authors suggested that both the tribe Paronychieae and the genus *Paronychia* are polyphyletic too. Furthermore, the circumscription of Paronychieae based on the indehiscence of fruits and reduced number of ovules cannot be correlated with the DNA sequence data of the *ndhF* region [3]. At the genus rank, *Paronychia* does not form a monophyletic group based on *ITS*, *matK*, and *rps16* data [5,6]. In particular, Oxelman et al. [6] showed that *Paronychia* is grouped in two different clades, with *Gymnocarpos* Forssk. and *Herniaria* L. as sister genera (the “Herniaria–Paronychia complex” *sensu* Fior et al. [5]). Note, however, that all these studies used a reduced number of samplings: eight and two *Paronychia* species were considered, respectively, by Fior et al. [5] and Oxelman et al. [6] with no taxa from South America. A wider study of *Paronychia* was carried out by Alvarez [7] who considered 15 species. The results obtained by Alvarez [7] confimed that *Paronychia* is polyphyletic in coherence with [5,6] and that *Paronychia* s.str., sister to *Gymnocarpos*, includes the type of the genus (*P. argentea* Lam.; see [8]) and North and South American species which form, in turn, a well-supported subclade. Additionally, [7,9] showed the polyphyly of *Paronychia*; Alvarez [7] highlighted that Andean *Paronychia* species form a monophyletic group. Relationships into the South American clade are still unresolved.

South America is the main center of diversity of the genus *Paronychia*, including 29 species (42 taxa including infraspecific ranks, i.e., subspecies, varieties, subvarieties, and forms) [7,9,10,11,12,13,14]. During the research we found three populations which cannot be identified with any of the currently accepted South American *Paronychia* species and, therefore, we decided to describe them as new species. So, the number of species occurring in South America reaches 30. According to the classification proposed by Chaudhri [11], the main stream of the American members of the genus (ca. 61 species) belong to the *P.* sect. *Paronychia*.

As part of the ongoing studies on South American Caryophyllaceae (e.g., [8,9,14,15,16,17,18,19,20]) which are, in turn, part of the Flora of Argentina project, ed. F. O. Zuloaga (by DI), and the *Caryophyllales Network*, ed. N. Korotkova (by DI and DBMT), we here present a nomenclatural synopsis of the *Paronychia* taxa occurring in South America, with some taxonomic notes.

## 2. Material and Methods

Field surveys were carried out in the highlands of the Central Andes since 2005; collected material was deposited at B, CUZ, F, HSP, HOXA, HSP, HUSA, HUT, K, MO, MOL, RO, and USM, and it was useful to carry out necessary morphological comparison among the various taxa, just to verify the preliminary taxonomic value of them. In addition, herbarium specimens were also examined. Herbaria checked were: B, CONC, COL, CUZ, E, F, FI, G, GH, GOET, HSP, HOXA, HSP, HUSA, HUT, K, LIL, LP, LPZ, MO, MOL, MOQ, MPU, NY, P, PRC, S, SGO, SI, U, US, USM, and WAG (acronyms according to Thiers [21]). Relevant literature (protologues included) and online databases were analyzed [10,11,12,22,23,24]. The macro-morphological features described are based on detailed field notes and photographs made from fresh and dry material. The characters (both vegetative and sexual) of the new taxon and the related species *Paronychia hieronymi* Pax were studied under an Olympus SZX10 stereo microscope and an NSZ-405 1X-4.5X stereo microscope, followed by the morphological analyses of the roots, stems, leaves, flowers, and fruits, both from freshly collected and dried herbarium material. The variability of the characters was examined by box plots, performed using the software NCSS 2007.

## 3. Results and Discussion

Among the 30 South American species of *Paronychia* (Table 1, Figure 1), Peru has the highest number of taxa (17), followed by Argentina (12) and Bolivia (10); the other countries include fewer than 10 taxa (Uruguay just one species).

### Taxonomic Treatment

Names of the taxa are numbered (Arabic number: 1, 2, etc.) and presented in alphabetical order. Infraspecific taxa are organized using letters (a, b, etc.), a second Arabic number after letters (a1, a2, etc.), and Roman numbers (I, II, etc.).

**1. *Paronychia arbuscula*** Gay Fl. Chil. 2(4): 520–521, 1846 [1847] ≡ *Spergularia arbuscula* (Gay) I.M. Johnst., Contr. Gray Herb. 85: 40. 1929.

Neotype (here designated): Chile, Coquimbo, *s.d.*, *C. Gay* (P00712710!, image available at https://plants.jstor.org/stable/viewer/10.5555/al.ap.specimen.p00712711, accessed on 19 February 2023).

*Notes.* Gay [25] gave a diagnosis and a detailed description for his *Paronychia arbuscula*; the provenance (“provincia de Coquimbo”) was also reported, but no syntype was cited. Four specimens collected by C. Gay in Chile were traced at F (barcode F0075448F, a fragment), GH (GH00013892) and P (P00712710, P00712711). Only one of them (P00712710) bears the precise locality as reported in the protologue (Coquimbo), the other exsiccata having labels with only “Chili” annotated. So, P00712710 would be the best candidate for the typification purpose. However, none of the mentioned specimens has the date of collection. Although Gay’s herbarium and types are preserved at P, whereas duplicates are in several other European and American herbaria (see [26]), by lacking the collection date, we cannot be sure that P00712710 was an ante-1846 addition to the collection and, therefore, we prefer to avoid it as a lectotype. According to Art. 9.8 of ICN, a neotypification is required. We here designate the specimen P00712710 as the neotype for the name *Paronychia arbuscula*. The type matches the diagnosis and corresponds to the current concept of the species (see e.g., [11]).

**2. *Paronychia arequipensis*** (Chaudhri) Montesinos & Iamonico, *comb. et stat. nov.* = *Paronychia microphylla* subsp. *microphylla* var. *arequipensis* Chaudhri, Revis. Paronychiinae: 151. 1968 (as “*arequepensis*”). Figure 2A.

Holotype: Peru, Arequipa, Arequipa, Cerrillo Lavandia, nr. Laspinas, N. of Arequipa, 2200 m, 15 September 1938, *W.J. Eyerdam & A.A. Beetle 22161* (GH!; isotypes: SI094815!, K000979328 (https://plants.jstor.org/stable/viewer/10.5555/al.ap.specimen.k000979328, accessed on 19 February 2023), MO492982 (https://plants.jstor.org/stable/viewer/10.5555/al.ap.specimen.MO492982, accessed on 19 February 2023), S (*non vidi fide* [11]), UC (*non vidi fide* [11])).

*Notes.* Chaudhri [11] validly published *Paronychia microphylla* subps. *arequepensis* var. *arequepensis* through a detailed diagnosis in which he explains “*This is a very characteristic species, distinguished by its villous stems, coriaceous, often sharply mucronate, ovate-lanceolate leaves, almost stellate flower-clusters and densely villous flowers*”. *P. microphylla* and *P. arequepensis* differ from each other by the habit (chamaephyte up to 30 cm tall in *P. microphylla* vs. caespitose phanerophyte up to 1.2 m in *P. arequepensis*), size of the leaves (4.00 × 1.50 mm vs. 8.00 × 2.25 mm in *P. arequipensis*), shape of the leaves (narrowly elliptic-oblong to lanceolate vs. narrowly ovate in *P. arequipensis*), and length of the apical awn of sepals (0.25–0.30 mm vs. 0.35–0.45 mm in *P. arequipensis*). In agreement with [11], this taxon is represented by an isolated population in south Peru. Different expeditions carried out by the first author (DBMT) have led to its finding in several parts of the Arequipa department, including populations found in the Moquegua department as studied by Chicalla-Rios [27] but no populations have been found further south towards the Tacna department and Arica in N Chile. We also verified that distinguished character-states of this taxon are constant in Peruvian populations [28] and, therefore, we here propose to treat Chaudhri’s variety at species rank, and correct the misspelled name from *arequepensis* to *arequipensis* since the epithet corresponds to the city of Arequipa (see Art. 60 of ICN [29]). The species are prone to declining, caused by the uncontrolled urbanization growth in the surroundings of the lower mountain slopes of the city of Arequipa where the species frequently occurs and it deserves critical reviews on its conservation status.

*Specimens examined*: PERU, Arequipa, Arequipa, along road Arequipa-Yura, 2680–3260 m, 13 April 2006, *H. van der Werff* et al. *20474* (MO6128130!); Arequipa, Arequipa, along road Arequipa-Yura, 3260 m, 13 April 2006, *H. van der Werff* et al. *20510* (MO6128129!); Arequipa, Arequipa, Yura, camino pedregozo y desértico en Yura, 2645 m, 20 April 2011, *D.B. Montesinos 3139a* (USM!); Arequipa, Arequipa, Characato, 2700–2900 m, 27 April 2000, *M. Weigend* et al. *2000/S24* (HUT400094!); Arequipa, Arequipa, Uchumayo, Capua, 2646 m, 12 March 2017, *D.B. Montesinos* et al. *5042* (B100745287!, O227292!); Arequipa, Arequipa, Uchumayo, Capua, 2616 m, 12 March 2017, *D.B. Montesinos* et al. *5037* (B100761249!); Moquegua, Mariscal Nieto, Torata, Pampas de Jaguay, 2420 m, 14 March 2017, *D.B. Montesinos* et al. *5083* (B100761248!); Moquegua, General Sánchez Cerro, Puquina, Road to Omate, 3036 m, 26 April 2017, *D.B. Montesinos & K. Chicalla 5355* (B100745250!); Arequipa, Arequipa, outside Mollebaya, towards the mountain Lapacheta Chica, 2457 m, 5–16 February 2003, *N. Dostert & F. Cáceres 1008* (B100208348!, B100208349!); Moquegua, General Sánchez Cerro, Omate, Road from Coalaque to Omate, 2482 m, 13 March 2017, *D.B. Montesinos* et al. *5065* (B100745284!, O227297!); Arequipa, Arequipa, zwischen Arequipa und Yura, 2500 m, 17 March 1972, *G. Mueller* et al. *1842b* (LPZ!). Arequipa, Arequipa, 12 km S of Arequipa, 2200 m, 14 September 1938, *W.J. Eyerdam & A.A. Beetle 22125* (SI094814!).

**3. *Paronychia bogotensis*** Triana & Planch. Ann. Sci. Nat. Bot., sér. 4, 17: 147. 1862.

Lectotype (designated here): Colombia, Andes de Bogotá, la Peña, alt. 2700 m, May 1855; *J. Triana s.n.* (COL000001167, image available at https://plants.jstor.org/stable/viewer/10.5555/al.ap.specimen.col000001167, accessed on 19 February 2023); isolectotypes: FI005352! (https://plants.jstor.org/stable/viewer/10.5555/al.ap.specimen.fi005352, accessed on 19 February 2023), K000486397! (https://plants.jstor.org/stable/viewer/10.5555/al.ap.specimen.k000486397, accessed on 19 February 2023), P00335870! (https://plants.jstor.org/stable/viewer/10.5555/al.ap.specimen.p00335870, accessed on 19 February 2023), and P00335871! (https://plants.jstor.org/stable/viewer/10.5555/al.ap.specimen.p00335871, accessed on 19 February 2023).

*Notes*. After the detailed description, Triana and Planchon [30] provided the provenance and two collectors (“Andes de Bogota, la Peña, alt. 2077 mètres, dans les endroits sablonneux (Tr.); Bogota (Goudot)”); these citations represent syntypes according to Art. 9.6 of ICN. We found seven specimens at COL (COL000001167), FI (FI005352), K (K000486397), MPU (MPU012438), and P (P00335870, P00335871, P00335872). P00335872 was collected by J. Goudot in 1844, MPU012438 was annotated with “*Paronychia bogotensis nob.*|*legit. Triana*”, whereas the other five specimens are part of the “HERBARIO DE J. TRIANA” (COL000001167; date of collection May 1855) or of the “Voyage de J. TRIANA 1851–1857” as reported in the original labels. By the comparison among these scripts, the label occurring on the specimen MPU012438 (https://plants.jstor.org/stable/viewer/10.5555/al.ap.specimen.mpu012438, accessed on 19 February 2023) was not annotated by J. Triana and we prefer to exclude this specimen as a lectotype. The other specimens are, instead, clearly part of the original material used by [30] to describe *Paronychia bogotensis*. The main set of Triana’s herbarium and type are currently preserved at COL [31]. Therefore, we here designate COL000001167 as the lectotype of the name *Paronychia bogotensis*; FI005352, K000486397, P00335870, and P00335871 are isolectotypes. The types match the diagnosis and correspond to the current concept of the species (see e.g., [11]).

*Specimens examined*: COLOMBIA, Bogota, près El rio del Ovispo, 1844, *J. Goudot 2* (P00335872!, syntype); Cundinamarca, Barrio San Cristobal, Bogotá, 2600 m, 10 December 1944, *F.R. Fosberg 22393* (B101156011!).

**4. *Paronychia brasiliana*** DC., Encycl. 5: 23. 1804 ≡ *Paronychia bonariensis* DC., Prodr. 3: 370. 1828, *nom. superfl. et illeg*.

Type (lectotype designated by Chaudhri [11] (pag. 152) as ”holotype”, here corrected according to Art. 9.10 of ICN): Argentina, Buenos Aires, Montevideo, November 1967, *Commerson s.n.* (G-DC, *non vidi fide* Chaudhri, 1968: 152); isolectotypes: MPU011320! (https://herbier.umontpellier.fr/zoomify/zoomify.php?fichier=MPU011320, accessed on 19 February 2023), P00712715! (http://mediaphoto.mnhn.fr/media/1443392961187PiUo29G0kl6VSkfC, accessed on 19 February 2023), P00712716! (https://plants.jstor.org/stable/viewer/10.5555/al.ap.specimen.p00712716, accessed on 19 February 2023), P00712717! (https://science.mnhn.fr/institution/mnhn/collection/p/item/p00712717?listIndex=3&listCount=15, accessed on 19 February 2023), P00712718! (https://plants.jstor.org/stable/viewer/10.5555/al.ap.specimen.p00712719?loggedin=true, accessed on 19 February 2023), P00712719! (https://plants.jstor.org/stable/viewer/10.5555/al.ap.specimen.p00712719?loggedin=true, accessed on 19 February 2023).

*Notes.* Chaudhri [11] stated “Type: Argentine: Buenos Aires (Bonaria), Commerson ix.1767, (holo. G-DC! iso! P!)”. However, Candolle [32] did not indicate any holotype in the protologue of *Paronychia brasiliana* (Art. 9.1 of ICN) but reported a syntype, i.e., “Cette plante a été trouvée par Commerson dans les chemins de Monte-Video … (*V. s.*) [=*vidi siccus*]”. According to Art. 9.10 of ICN, Chaudhri’s use of “holo” (=holotype) is an error to be corrected to lectotype. The indication of the P specimen as an isotype (“iso”) should be corrected to isolectotype. Furthermore, we traced five pertinent specimens at P (P00712715, P00712716, P00712717, P00712718, P00712719), all part of Poiret’s collection and collected by P. Commerson in Montevideo/Buenos Aires. So, all these five specimens are isolectotypes for the name *Paronychia brasiliana*. Finally, a further isolectotype was found at MPU (MPU011320). All the original material found matches the diagnosis and corresponds to the current concept of the species (see e.g., [11]).

*Paronychia bonariensis* was published by Candolle [33] in the 3th volume of his *Prodromus* where, after the description, the provenance (“in Bonariâ, et Monte-Video?”), the collector (“Commerson”) and the citation “Paronychia Brasiliana DC. in Lam. dict. 5 p. 23” were given. This latter statement refers to the 5th volume of Lamarck’s (1804) *Encyclodepiè Botanique* where *P. brasiliana* was validly published. So, Candolle [33] provided a new name for the legitimate *P. brasiliana*. Hence, the name *P. bonariensis* is superfluous and illegitimate according to Art. 52.1 of ICN. Note that [33] also listed *P. bonariensis* as a synonym of *P. brasiliana*.

**4a. *Paronychia brasiliana*** DC., Encycl. 5: 23. 1804 subsp. ***brasiliana***.

**4b. *Paronychia brasiliana*** DC., Encycl. 5: 23. 1804 subsp. ***pubescens*** (Chaudhri) Iamonico & Montesinos *stat. nov.* ≡ *Paronychia brasiliana* subsp. *brasiliana* var. *pubescens* Chaudhri, Meded. Bot. Mus. Herb. Rijks Univ. Utrecht 285: 152. 1968.

Lectotype (here designated): Argentina, Buenos Aires, Monte Hermoso, 1916, *Carette s.n.* (LP003078!, https://plants.jstor.org/stable/viewer/10.5555/al.ap.specimen.lp003078, accessed on 19 February 2023); isolectotype: LP006785 (https://plants.jstor.org/stable/viewer/10.5555/al.ap.specimen.lp006785, accessed on 19 February 2023).

*Notes.* Chaudhri [11] described this variety providing a short diagnosis (“caulium ramificationibus retrorse pubescentibus distinguenda”); the following type indication was given: “Type: Argentine: Prov. Buenos Aires, Monte Hermoso, /1916, *E. Carette* (LP)”. We traced two specimens at LP (LP003078 and LP006785) both collected by E. Carette in 1916 in Monte Hermoso. According to Art. 40.2 Note 1, since these two collections are clearly part of the same gathering, the name *Paronychia brasiliana* subsp. *brasiliana* var. *pubescens* was validly published and Chaudhri’s citations are syntypes (see also Art. 9.6 of ICN). As a consequence, a lectotype should be designated (Art. 9.3 of ICN). We here designate LP003078 as the lectotype of the varietal name since it appears to be better preserved and includes more inflorescences whose features are important in the taxonomy of the genus *Paronychia* (see e.g., [11]). LP006785 is the isolectotype. These two LP specimens match the diagnosis and correspond to the current concept of the species (see e.g., [11]).

Taxa *brasiliana* and *pubescens* have different and not overlapping distribution areas, i.e., respectively in Argentina (Córdoba Province), South Africa, and Australia [34] and Buenos Aires Province in Argentina. So, we here propose to treat these two taxa at the subspecies rank of *Paronychia brasiliana.*

**5. *Paronychia cabrerae*** Chaudhri, Meded. Bot. Mus. Herb. Rijks Univ. Utrecht 285: 181, 1968.

Holotype: Argentine: Prov. Jujuy: Dept. Humahuaca, Azul Pampa, c. 3700 m, 17 February 1963, *A. L. Cabrera* et al. *15210* (LP003079!, https://plants.jstor.org/stable/viewer/10.5555/al.ap.specimen.lp003079, accessed on 19 February 2023).

*Specimens examined*: ARGENTINA, Prov. de Jujuy, Dep. Humahuaca, Mina Aguilar, Espinazo del Diablo, 3800 m, *A.L. Cabrera* et al. *18992* (LP!); Dep. Capital, entre León y Nevado de Chañi, Las Cuevas, 3000 m, March 1963, *H.A. Fabris* et al. *4055* (LP!).

**6. *Paronychia camphorosmoides*** Cambess. Flora Brasiliae Meridionalis (quarto ed.) 2(15): 187–188. 1830.

Lectotype (designated by Chaudhri [11], first step), second-step lectotypification (Art. 9.17 of ICN) here designated): Brazil: Prov. de Saint Paul (Sao Paulo), 1816–1821, *A. de Saint-Hilaire, Catal. C2 1513* (P00712722, https://plants.jstor.org/stable/viewer/10.5555/al.ap.specimen.p00712722?loggedin=true, accessed on 19 February 2023); isolectotype: P00712723 (https://science.mnhn.fr/institution/mnhn/collection/p/item/p00712723?listIndex=2&listCount=10, accessed on 19 February 2023).

*Note.* Cambessèdes [34] gave a diagnosis, a detailed description, and the provenance (“Prope *Egreja Velha* in part provinciae S. Pauli dictâ *Campo Gereaës*”). According to Art. 9.10 of ICN, [11] the use of “holo” (=holotype), for a P specimen collected by A. de Saint-Hilare (no. 1513) is an error and it should be corrected to lectotype. Note, however, that there are two specimens at P (P00712722 and P00712723) bearing the no. 1513: P00712723 bears a complete label with locality and date of collection, whereas the label of P00712722 just reports “*Paronychia camphorosmoides Cambess.*”. However, the occurrence of a small label with “*1513*” on the exsiccatum of P00712722 led us to consider it as part of the original material. So, Chaudhri’s designation can be considered as the first-step typification and a second step is required by Art. 9.17 of ICN. P00712723 is an isolectotype. The types match the diagnosis and correspond to the current concept of the species (see e.g., [11]).

*Specimens examined*: BRAZIL, Brasilia, RGS, Serra de Rocinha, P. Bom Jesus, 3 February 1952, *B. Rambo 53884* (B101156012!, B101114296!); S. Paulo, Villa Emma, December 1933, *Brado 13073* (B101114302!); RGS, Serra da Rocinha, P. Bom Jesus, 18 January 1950, *B. Rambo 45413* (B101114299!); RGS, Passo do Socorro, P. Vacaria, 26 December 1951, *B. Rambo 51466* (B101114298!).

**7. *Paronychia chilensis*** DC., Prodr. 3: 370. 1828. Figure 2B.

Type (lectotype designated by Chaudhri [11] (pag. 171) as ”holotype”, here corrected according to Art. 9.10 of ICN): Chile, Región del Biobío. “… circà Conceptionis urbem…”, *d’Urville s.n.* (G-DC, *non vidi fide* [11]); isolectotype: P00712724! (https://plants.jstor.org/stable/viewer/10.5555/al.ap.specimen.p00712724?loggedin=true, accessed on 19 February 2023).

*Notes. Paronychia chilensis* was validly described by Candolle [33] by a diagnosis; the provenance and collector (“Conceptionis urben in Chili, legit cl. [clarissimo] D’Urville”) were also provided; moreover “v. s.” (=*vidi siccus*) was also reported, indicating that A. P. Candolle had seen at least one specimen. Candolle’s citation of provenance and collector can be considered as a syntype according to Art. 9.6 of ICN. Chaudhri [11] wrongly reported “holo. G-DC!” for a specimen collected in Conceptión by D’Urville. In fact, as stated above, [33] did not indicated any holotype for *P. chilensis* (Art. 9.1 of ICN). According to Art. 9.10 of ICN, Chaudhri’s use of “holo” (=holotype) is an error to be corrected to lectotype. Further original material was traced at P (P00712724) and it bears some pieces of a single plant collected by D’Urville in Conceptión, as reported on the original label (bottom-left corner of the sheet). This P specimen is an isolectotype. Finally, in the JSTOR database, there are two specimens (MA811550 and MA811551) indicated as “original material” (see https://plants.jstor.org/stable/viewer/10.5555/al.ap.specimen.ma811550, accessed on 19 February 2023 and https://plants.jstor.org/stable/viewer/10.5555/al.ap.specimen.ma811551, accessed on 19 February 2023). However, these two MA specimens cannot be considered for the lectotypification purpose, being part of the Ruiz and Pavon’s Herbarium Peruvianum and no reference to D’Urville or Conceptión (Chile) is reported in the label.

Chaudhri [11] recognized two subspecies, i.e., *Paronychia chilensis* subsp. *chilensis* and *P. chilensis* subsp. *subandina* (Phil.). Chaudhri based them on stem hairiness, style length, and stigma structure; *P. chilensis* subsp. *chilensis* was further classified into two varieties, i.e., *P. chilensis* subsp. *chilensis* var. *chilensis* and *P. chilensis* subsp. *chilensis* var. *mutica* (Phil) Reiche, based on hairiness and shape of the leaves. The phenotypic variability of *P. chilensis* is quite complicated and, based on our observations of herbarium specimens, we prefer to avoid a taxonomic conclusion about these taxa. So, for the moment, we accept Chaudhri’s classification, awaiting further investigations.

*Specimens examined*: ARGENTINA, Prov. Buenos Aires, Pdo. Coronel Suarez, Abra del Pantanoso Viejo, 14 December 1979, *L.A. Pertusi 149* (LP!); Buenos Aires, Pdo. Tornquist, Cerro de la Ventana, Abra de la Ventana, 30 November 1978, *Proyecto Ventania 472* (LP!); Dihuel Calel, Sierras de Lihuel Calel, 25 October 1978, *Steibel et al. 5977* (LP!); Misiones, 29 December 1907, *E.L. Elkmann 1874* (MO1034604!); Prov. de Buenos Aires, Campo de Mayo, October 1936, *G. Valencia s.n.* (MO1223763!); BOLIVIA, Ex. Herbario Collegii Columbiae, a N.L. Britton et H.H. Rusby distributae, *M. Bang 1970* (MO1766133!); La Paz, 3800 m, 25 March 1913, *O. Buchtien 546* (MO965291!); BRASIL, Paraná, Guarapuava, 50 km ao oeste de Guarapuava, 1000 m, 15 December 1965, *Reitz & Klein 17715* (MO1829482!); ECUADOR, Pichincha, borde de camino, 2800 m, 7 February 1928, *C. Fermin 363* (US1420591!); Azuay, Cuenca, 8200–8900 ft., 14 April 1945, *W.H. Camp E-2653* (NY!); Loja, 2250 m, 2 May 1946, *R. Espinosa 267* (F1197123!); Loja, cerro Pucará, 2300 m, 18 May 1946, *R. Espinosa 409* (F11977113!); PARAGUAY, *P. Jörgensen 4310* (MO971600!); PERU, Lima, Huarochirí, San Bartolomé, Monte de Zárate, 1400–3550 m, 24–26 April 2009, *P. Gonzáles* et al. *611* (USM256844!); Lima, Yauyos, Ajacki, cerro al este de Tupe, 2850 m, 4 January 1952, *E. Cerrate 1038* (USM18456!); Lima, Canta, Lachaqui, canal de la Toma, 3650 m, 17 May 1998, *G. Vilcapoma 4795* (USM263250!); Huánuco, Acomayo, Chinchao, alturas de Micho, 2500 m, February 1940, *C.A. Ridoutt s.n. 11528*(?) (USM18451!); Cajamarca, Contumazá, alrededores de guzmando, 2700 m, 2 April 1981, *A. Sagastegui* et al. *9662* (HUT16050!); Amazonas, Bongará, 2 km below Campamento Ingenio, along Rio Utcubamba, 1250–1275 m, 28 January 1964, *P.C. Hutchinson & J. Kenneth 3843* (K!, US2467464!).

**7a. *Paronychia chilensis*** DC. subsp. ***chilensis*** var. ***chilensis***.

**7b. *Paronychia chilensis*** DC. subsp. ***subandina*** (Phil.) Chaudhri, Meded. Bot. Mus. Herb. Rijks Univ. Utrecht 285: 172, 1968 ≡ *Paronychia subandina* Phil., Annal. Univ. chil. 85: 323. 1894.

Lectotype (designated by Chaudhri [11] as “holotype”, here corrected according to Art. 9.10 of ICN): Chile, Dept. Ovalle, in valle fluminis Torca, 1889–90, *G. Geisse* (SGO000001976!, image available at https://plants.jstor.org/stable/viewer/10.5555/al.ap.specimen.sgo000001976, accessed on 19 February 2023; isolectotypes: SGO000001977! and SGO000001978!).

*Note.* Chaudhri [11] reported “holo. SGO. 48927!” for a specimen collected in in *valle fluminis Torca*. However, Philippi [35] did not indicate any holotype for *P. subandina* (Art. 9.1 of ICN). According to Art. 9.10 of ICN, Chaudhri’s use of “holo” (=holotype) is an error to be corrected to lectotype.

**7a1. *Paronychia chilensis*** DC. subsp. ***subandina*** var. ***subandina***.

**7a2. *Paronychia chilensis*** DC. subsp. ***subandina*** var. ***mutica*** (Phil.) Reiche ≡ *Paronychia mutica* Phil., Linnaea 33: 79. 1864.

Lectotype (designated by Chaudhri [11] as “holotype”, here corrected according to Art. 9.10 of ICN): Argentina, Aconcagua, prope Los Molles, November 1862, *L. Landbeck 772* (SGO000001974, image available at (https://plants.jstor.org/stable/viewer/10.5555/al.ap.specimen.sgo000001974, accessed on 19 February 2023); isolectotype: SGO000001975, image available at https://plants.jstor.org/stable/viewer/10.5555/al.ap.specimen.sgo000001975, accessed on 19 February 2023.

*Notes on Paronychia mutica.* Philippi [35] provided a description and the provenance (“Prope los Molles in parte litorali prov. Aconcagnae invenit orn. Landbeck”). In addition, [11] indicated a specimen deposited at SGO (barcode SGO000001974) as the holotype. According to Art. 9.10 of ICN [29], Chaudhri’s use of “holo” (=holotype) is an error and it should be corrected to lectotype. We also traced an isolectotype (SGO000001975).

**8. *Paronychia coquimbensis*** Gay, Fl. Chil. 2(4): 521–522. 1846 [1847].

Lectotype (here designated): Chile, 1838, *Gay s.n.* (P00712726, https://plants.jstor.org/stable/viewer/10.5555/al.ap.specimen.p00712726, accessed on 19 February 2023).

*Notes*. Gay [25] gave a diagnosis, a detailed description, and the provenance (“departamento de Coquimbo y especialmente en el camino de la Serena á Arqueros”) for his *Paronychia coquimbensis*. Five specimens were found at K (K000486388 and K000486389) and P (P00712726, P00712727, and P00712728). K000486388, K000486389, and P00712728 each bear an original label without the date of collection and, therefore, we cannot be sure that they are ante-1846 additions to the collection and part of the original material for *P. coquimbensis*. The other traced specimens (P00712726, P00712727, and P00712728), all collected by C. Gay in Chile, in 1838 and 1833, respectively, are eligible as lectotypes. Since P00712726 is represented by a larger exsiccatum than P00712727 and P00712728, also including many flowers, whose features are important in the identification of *Paronychia* species, we here designate P00712726 as the lectotype of the name *P. coquimbensis*. The type matches the diagnosis and corresponds to the current concept of the species (see e.g., [11]). Note that P00712727 and P00712728 each bear a printed “ISOTYPE” label which, however, has no effect on the lectotypification never being published on the basis of our literature analysis. Finally, the word “*Cotypus*” (reported in a separate label of P00712728), which is not formally regulated by the *Shenzhen Code*, is an obsolete term meaning syntype (or sometimes isotype or paratype; see [29]) and again it does not affect our proposed lectotypification and a second step is required by Art. 9.17 of ICN.

**8a. *Paronychia coquimbensis*** Gay subsp. ***coquimbensis*** var. ***coquimbensis***.

**8b. *Paronychia coquimbensis*** Gay subsp. ***coquimbensis*** var. ***appressa*** (Phil.) Chaudhri, Meded. Bot. Mus. Herb. Rijks Univ. Utrecht 285: 174. 1968 ≡ *Paronychia appressa* Phil., Linnaea 33: 79–80. 1864.

Lectotype (designated by Chaudhri [11] as ”holotype”, here corrected according to Art. 9.10 of ICN): Central Chile, Prov. Coquimbo, nr. Illapel, December 1862, *L. Landbeck 771* (SGO000001971, image available at (https://plants.jstor.org/stable/viewer/10.5555/al.ap.specimen.sgo000001971, accessed on 19 February 2023); isolectotype SGO000001970, image available at https://plants.jstor.org/stable/viewer/10.5555/al.ap.specimen.sgo000001970?loggedin=true, accessed on 19 February 2023.

*Notes on Paronychia appressa.* Philippi [35] provided a description and the provenance (“Prope Illapel crescit, unde specimina attulit orn. Landbeck”). Additionally, [11] indicated a specimen deposited at SGO (SGO000001971) as the holotype. According to Art. 9.10 of ICN, Chaudhri’s use of “holo” (=holotype) is an error and it should be corrected to lectotype. We also traced an isolectotype (SGO000001970).

**9. *Paronychia communis*** Cambess., Fl. Bras. Merid. (quarto ed.) 2(15): 186, 1829 [1830].

Lectotype (designated by Chaudhri [11], first step), second-step lectotypification (Art. 9.17) here designated: Brasil, *São Borja*, Sao Paulo, St. Hilaire in 1816-21 voy., 1511 (& 2030bis) (P00712713!, image available at https://plants.jstor.org/stable/viewer/10.5555/al.ap.specimen.p00712713, accessed on 19 February 2023).

*Notes.* Cambessèdes [34] published *Paronychia communis* by a diagnosis and a detailed description; the proveannce (“… provinciae S. Pauli dictâ *Campo Geraës* … provinciarum S. Catharinae, *Rio Grande de S. Pedro do Sul*, Cisplatinae, et Missionum”) was also given. Chaudhri [11] cited a specimen collected by *St. Hilare* in Brazil (deposited at P) as “lecto.-holo.”. This strange statement was used by [11] two times in his *Revision of the Paronychiinae* (the second one for *P. chionaea* Boiss. subsp. *chionaea* var. *chionaea*, on page 238). So, we think that “lecto.-holo.” is not a typographical error, but would represent Chaudhri’s doubt in type designation. In the case of *P. communis*, the specimen cited by [11] was traced (P00335873) and it bears two plants associated with two collection numbers (1511 and 2030bis) as also indicated by [11]; two labels also occur on the sheet, i.e., “*Paronychia communis F. B.*|*Herb. Bresil.*|*St. Hilaire*” (linked with plant no. 1511, on the left of the sheet) and “*Brésil—Sao Paulo, campos Geraes*” (linked with plant no. 2030 bis, on the right of the sheet). Note that Art. 8.2 Note 1 states “… collecting numbers … alone do not necessarily denote different gatherings”. However, the next article of the *Shenzhen Code* (Art. 8.3) reports that “Multiple preparations from a single gathering that are not clearly labelled as being part of a single specimen are duplicates”. In our case, the occurrence of two plants (“preparations” according to Art. 8.3) and two labels does not lead us to consider them as surely part of a single gathering. Therefore, Chaudhri’s type indication (holotype or lectotype) is not correct in our opinion. According to Art. 9.19 of ICN, a lectotypification is necessary. Note that we traced a further two specimens which are part of the original material for *Paronychia communis*, i.e., P00712713 and P00335874 (https://plants.jstor.org/stable/viewer/10.5555/al.ap.specimen.p00335874?loggedin=true, accessed on 19 February 2023) both refer to plants collected by St. Hilare during his trip to Rio Grande do Sul between 1816 and 1821 (as reported on the original labels); both these specimens have the same collection number, i.e., “*2659bis*”. We here designate the specimen P00335873 as the lectotype of the name *P. communis* since it bears a further handwritten label (on the bottom-left corner of the sheet) including the generic name “*Paronychia*” (without a specific epithet), a description, and the locality of collection (“*In pascuis prope … S. Francisco de Borja*” (São Borja), a city in the north-west of the Brazilian state of Rio Grande do Sul). The type matches the diagnosis and corresponds to the current concept of the species (see e.g., [11]).

Chaudhri [11] recognized three varieties under *Paronychia communis* subsp. *communis*, i.e., *P. communis* subsp. *communis* var. *communis*, *P. communis* subsp. *communis* var. *chicligastensis* Chaudhri, and *P. communis* subsp. *communis* var. *pungentifolia* Chaudhri, based on leaf shape, flower length, and length of sepal awn; the former variety *communis* was further classified into two forms, i.e., *P. communis* subsp. *communis* var. *communis* f. *communis* and *P. communis* subsp. *communis* var. *communis* f. *subglabra* Chodat & Hassl. based on leaf hairiness. The phenotypic variability of *P. communis* is quite complicated and, based on our observations of herbarium specimens, we prefer to avoid a taxonomic conclusion about these taxa. So, for the moment, we accept Chaudhri’s classification, awaiting further investigations.

*Specimens examined*: BOLIVIA, Tarija, Arce, valley of the Rio Chillaguatas, below Rancho Nogalar on train between Sidras and Tariquia, 1100 m, 14–16 October 1983, *J.C. Solomon 11626* (MO3396635!); BRASIL, Paraná, Sierra do Mar, Piranga, 830 m, 23 August 1914, *P. Dusén 835a* (MO905629!); ECUADOR, Quito, vicinity of Quito, 1 November 1918, *J.N. Rose & G. Rose 23541* (NY!); Pichincha, Quito, Panecillo, 2900 m, 16 May 1939, *E. Asplund 6062* (NY47319!, MO5733659!); Azuay, Cuenca, 2550 m, 25 September 1955, *E. Asplund 17795* (NY!); PARAGUAY, departamento Cordillera, Caacupe, 9 February 1984, *W. Hahn 2010* (MO3219560!); PERU, Ancash, Huaraz, Huascarán National Partk, quebrada Shallap, below lake, 3690–4100 m, 20 February 1985, *D.N. Smith* et al. *9647* (USM62802!); Ancash, Huaraz, Huascarán National Partk, quebrada Ishinca, N side of valley, 4400 m, 13 February 1985, *D.N. Smith* et al. *9523* (USM67571!); Peru, Cajamarca, Cajamarca, surroundings of Cajamarca, northern Peru, 2700–3700 m, 15 May 1986, *B. Becker & F.M. Terrones 1067* (LPZ!); Cajamarca, Cajamarca, zum Cumbe Mayo, 3000 m, 7 January 1979, *G. Müller & P. Gutte 8741ab* (LPZ!); Cajamarca, Cajamarca, nach Celendín, 3400 m, 12 January 1979, *G. Müller & P. Gutte 9365a* (LPZ!); Cajamarca, Cajamarca, surroundings of Cajamarca, northern Peru, 2700–3700 m, 6 February 1986, *B. Becker & F.M. Terrones 359* (LPZ); Cajamarca, Cajamarca, 2800 m, 11 January 1979, *G. Müller & P. Gutte 9081ab* (LPZ!).

**9a. *Paronychia communis*** subsp. ***communis*** var. ***communis***.

**9a1. *Paronychia communis*** subsp. ***communis*** var. ***communis*** f. ***communis***.

**9a2. *Paronychia communis*** subsp. ***communis*** var. ***communis*** f. ***subglabra*** Chodat & Hassl., Bull, Herb. Boiss. ser. 2, 3: 791. 1903.

Type: not designated.

*Specimens examined*: COLOMBIA, Cundinamarca, Chapinero near Bogotá, May 1923, *H. Pring 59* (MO904843!).

**9b. *Paronychia communis*** Cambess. subsp. ***communis*** var. ***chicligastensis*** Chaudhri, Meded. Bot. Mus. Herb. Rijks Univ. Utrecht 285: 165. 1968.

Holotype: ARGENTINA, Provincia: Tucumán, Dpto. Chicligasta, La Cascada, 14 June 1948, *T. Meyer 14120* (LIL000448!, image available at https://plants.jstor.org/stable/viewer/10.5555/al.ap.specimen.lil000448, accessed on 19 February 2023).

**9c. *Paronychia communis*** Cambess. subsp. ***communis*** var. ***pungentifolia*** Chaudhri, Meded. Bot. Mus. Herb. Rijks Univ. Utrecht 285: 165. 1968.

Holotype: Brazil, Brasilia, RGS, 17 October 1949, *B. Rambo 43913* (LIL000449!, image available at https://plants.jstor.org/stable/viewer/10.5555/al.ap.specimen.lil000449, accessed on 19 February 2023).

*Specimens examined*: ARGENTINA, provincia de Buenos Aires, Tandil, 300 m, 2 March 1946, *A. Krapovickas 2977* (MO1305846!); BRASIL, Rio G. do Sul, Iacolumí p. Grabataí, 11 January 1950, *B. Rambo 45246* (MO1629256!); Brasilia, Rio Grande do Sul, Bom Jesus Pezenda Bernardo Velho, 16 January 1947, *B. Rambo 34715* (MO1591298!).

**10. *Paronychia compacta*** Montesinos & Iamonico, *nom. nov. pro Paroyichia andina* Phil., Anales Univ. Chile 36: 172–173. 1870, *nom. illeg.* (later homonym; Art. 53.1 of ICN) non Gray 1854 (Figure 2C).

Type (lectotype designated by Chaudhri [11] as ”holotype”, here corrected according to Art. 9.10 of ICN): Perú. [Herbarium of the U. S. Exploring Expedition under the commend of Capt. Wilkes] Andes, Alpamarca, 1838–1842, *s.coll. s.n.* (NY00342590!, https://plants.jstor.org/stable/viewer/10.5555/al.ap.specimen.ny00342590, accessed on 19 February 2023).

*Etymology.* The specific epithet refers to the compactness of the plants, which is a unique habit feature amongst other *Paronychia* across South America.

*Notes. Paronychia andina* was validly described by Philippi [36] who provided both a diagnosis and a detailed description; the provenance (“Hab. Baños; Casa Cancha to Culnai and Alpamarca, in the high Andes of Peru (also gathered in the same region by Mr. M’Lean, and at Cerro Pasco by Mr. Matthews)”) was also given. These citations represent syntypes according to Art. 9.6 of ICN. Despite this, Chaudhri [11] indicated a specimen deposited at SGO (SGO000001969) as the holotype. According to Art. 9.10 of ICN, Chaudhri’s use of “holo” (=holotype) is an error and it should be corrected to lectotype. However, [11] chose the following specimen as type: “Andes inter Mendoza et Chile, Portezuelo del Portillo, Mendoza. Reed, i.1870”. Since this specimen was collected in Chile, but the protologue of *P. andina* reports localities from Peru, SGO exsiccatum cannot be considered as part of the original material and, as a consequence, is not usable as a lectotype (Arts. 9.3 and 9.4 of ICN). We traced the following relevant specimens:

➢K000486393: two pieces of a same plant collected by McLean in “*Perú*” (no specific locality was annotated); image at https://plants.jstor.org/stable/viewer/10.5555/al.ap.specimen.k000486393, accessed on 19 February 2023;➢K000486394: two pieces of a same plant collected in “Peru” and part of the Charles Wilkes’ UNITED STATES EXPLORING EXPEDITION (years 1838–1842) as reported on the printed label (same sheet of K000486393);➢K000486396: three pieces of a same plant collected by Pearce in “*Cerro del Pasco*|*2/4/63* [1863]” (same sheet of K000486393);➢GH00037820 (https://plants.jstor.org/stable/viewer/10.5555/al.ap.specimen.gh00037820, accessed on 19 February 2023): four pieces of a same plant collected in “*Baños Andes*|Perú” and part of the Charles Wilkes’ UNITED STATES EXPLORING EXPEDITION (years 1838–1842);➢NY00342590 (https://plants.jstor.org/stable/viewer/10.5555/al.ap.specimen.ny00342590, accessed on 19 February 2023): two pieces of a same plant collected in “*Alpamarca Andes of Peru*” and part of the Charles Wilkes’ UNITED STATES EXPLORING EXPEDITION (years 1838–1842);➢P00335877 (https://plants.jstor.org/stable/viewer/10.5555/al.ap.specimen.p00335877, accessed on 19 February 2023): two pieces of a same plant collected in “Perú” (no specific locality was annotated); this specimens is part of the Charles Wilkes’ UNITED STATES EXPLORING EXPEDITION (years 1838–1842);➢US00103386 (https://plants.jstor.org/stable/viewer/10.5555/al.ap.specimen.us00103386, accessed on 19 February 2023): two pieces of a same plant collected in “*Baños Perú & Casa Cancha*” (no collector was reported); this specimen is part of the Charles Wilkes’ UNITED STATES EXPLORING EXPEDITION (years 1838–1842) as reported on the printed label on the bottom-left corner of the sheet.

All the specimens listed above were originally annotated as “*Paronychia andina n.* sp.” and they can be considered as part of the original material for Philippi’s name (note that it is a later homonym of *P. andina* A. Gray [37] and, therefore, illegitimate under Art. 53.1 of ICN). NY00342590 appears to be the better-preserved specimen and it is here designated as the lectotype of Philippi’s name *Paronychia andina*. The type matches the diagnosis and corresponds to the current concept of the species (see e.g., [11]).

*Specimens examined*. PERU, Cusco, Urubamba, entre la quebrada de Huayoccari y las lagunas de Yanacocha y Kellococha, hacia las laderas Sureste de las lagunas, 2900–3600 m, 19–23 June 1989, *A. Tupayachi 1100* (MO3663854!); Moquegua, General Sánchez Cerro, Ichuña, Yanapuquio, ladera pedregoza, subnival, 4660 m, 17 April 2012, *D.B. Montesinos 3856* (USM!); Apurimac, Cotabambas, Haquira, Wiqapunku, rocky canyon with dense vegetation, 4234 m, 30 March 2017, *D.B. Montesinos & D. Cornejo 5264* (B100761301!); Lima, Yauyos, Huacracocha, debajo de la Laguna a más o menos 16 km de Tupe, 4300 m, 22 January 1952, *E. Cerrate 1235* (USM18912!); Ancash, Huari, Huascarán National Park, km from Cauish Tunnel, 4350 m, 30 March 1985, *D.N. Smith & F. Escalona 10158* (MO3314452!); Junin, Tarma, Puna de Tarma, 3900 m, 17 October 1951, *E. Cerrate 986* (USM18448!); Lima, Huarochiri, Ticllo, 5 March 1966, *E. Cerrate 4289* (USM18447!); Ayacucho, Lucanas, NE del campamento de Pampa Galeras, 4100 m, 22 May 1969, *O. Tovar 6256* (USM43587!); Cusco, Chumbivilcas, Uchucarco, 4431 m, 19 July 2007, *M. Morales* et al. *694* (USM253697!); Huancavelica, Huancavelica, arriba de Machajuay, entre Consica y Tinyajlla, 4000–4500 m, 29 March 1952, *O. Tovar 880* (USM90345!); *A. Weberbauer 2702* (B101156359!); *A. Weberbauer 984* (B101114297!); Peru, Apurimac, Cotabambas, Haquira, NW of Haquira, 4525 m, 28 March 2017, *D.B. Montesinos 5199* (B100761501!); Moquegua, General Sánchez Cerro, Ubinas, Matazo-Pillone road, 4598 m, 13 February 2016, *D.B. Montesinos* et al. *4489* (B100761307!); Arequipa, Caylloma, Sibayo, road to Condoroma, 4780 m, 16 February 2016, *D.B. Montesinos & S. von Mering 4515* (B100761284!); Arequipa, Caylloma, Sibayo, road to Condoroma, 4780 m, 16 February 2016, *D.B. Montesinos & S. von Mering 4508* (B100761162!); Apurimac, Cotabambas, Haquira, Wiqapunku, 4234 m, 30 March 2017, *D.B. Montesinos & D. Cornejo 5262* (B100761300!); Apurimac, Cotabambas, Haquira, Suitururi, 4220 m, 30 March 2017, *D.B. Montesinos & D. Cornejo 5243* (B100843040!); Junin, La Oroya, Pachacayo, Hda Cocha, 4320 m, 6 may 1974, *P. Gutte 3127b* (LPZ!); Cajamarca, 31 May 1986, *B. Becker & F.M. Terrones 1317* (LPZ!); Junin, Yauli, Ticlio, aus Huascacocha, 4800 m, 8 may 1971, *Müller* et al. *20b* (LPZ!); Junin, La Oroya, Morococha, Hacienda Pucará, 4700 m, 23 May 1974, *P. Gutte 2350b* (LPZ!); Junin, Yauli, Ticlio, aus Huascacocha, 4800 m, 8 May 1971, *Müller* et al. *20b* (LPZ!); Junin, La Oroya, zwischen Morococha, 4700 m, 24 May 1974, *P. Gutte 2404b* (LPZ!); Junin, La Oroya, Pachacayo, 35 km west oder Hda Cocha, 4700 m, 7 May 1974, *P. Gutte 3128b* (LPZ!).

**10a. *Paronychia compacta*** Montesinos & Iamonico subsp. ***compacta***.

**10b. *Paronychia compacta*** subsp. ***boliviana*** (Chaudri) Montesinos & Iamonico *comb. nov.* ≡ *P. andina* subsp. *boliviana* Chaudhri, Revis. Paronychiinae 192–193. 1968.

Holotype: Bolivia, La Paz, Chacaltaya, 30 km from La Paz, 4800 m, December 1934, *O. Buchtien 9392* (GH00037821!, image available at https://plants.jstor.org/stable/viewer/10.5555/al.ap.specimen.gh00037821, accessed on 19 February 2023); isotypes: (E, image available at https://plants.jstor.org/stable/viewer/10.5555/al.ap.specimen.gh00037821, accessed on 19 February 2023).

*Specimens examined*. BOLIVIA, above Copacabana, Lake Titicaca, on mountain top, 14,500 ft, 2 February 1903, *A.W. Hill 418* (K!); Bolivia, Khuri-Cuevas, 12 February 1928, *Troll 1454* (B!); Cochabamba, Tiraque, 20–25 km from main Cochabamba to Chapare highway on old road to Chapare, 4000 m, 18 April 1998, *J.R.I. Wood 135239* (K!); Cochabamba, Arani, 9.5 km by road SE Rodeo, then 2.5 km W, on road to Entel antenna, 3725 m, 29 March 1984, *G. Schmitt & D. Schmitt 91* (MO3272214!); La Paz, Murillo, 2.7 km SW of pass at head of Valle del zongo, 4600 m, 15 March 1984, *J.C. Solomon* et al. *11807* (MO3396522!); La Paz, Murillo, 3.4 km N of Milluni on road to Zongo, 4600 m, 25 April 1984, *J.C. Solomon & M. Moraes 13446* (MO04915128!); La Paz, B. Saavedra, Cotopampa, Cerro Pinita, 4530 m, 9 January 1983, *X. Menhofer X-1786* (LPZ!); La Paz, B. Saavedra, más arriba de Khata, 4100 m, 6 January 1983, *X. Menhofer X-1756* (LPZ!); Apolobamba-Kordillere Hochfläche von Ulla Ulla, 4450 m, 30 April 1984, *X. Menhofer X-2300* (LPZ!); Apolobamba-Kordillere, Aufstieg von der Estancia Canhuma zum Cerro Laramani (Canton Ulla Ulla), 4600 m, 22 January 1983, *X. Menhofer X-1903* (LPZ!); Khuri-Cuevas, 12 February 1928, *Troll 1454* (B101113871!).

**10c. *Paronychia compacta*** subsp. ***purpurea*** (Chaudri) Montesinos & Iamonico *comb. nov.* ≡ *P. andina* subsp. *purpurea* Chaudhri, Revis. Paronychiinae 193. 1968 (Figure 2D).

Holotype: Peru: Yanashallas, 35 km W of Huallanca, 78.25° W, 8.3° S, 5300 m, 2 October 1922, *Macbride & Featherstone 2480* (G00227055!, image available at https://plants.jstor.org/stable/viewer/10.5555/al.ap.specimen.g00227055, accessed on 19 February 2023.

*Specimens examined*. PERU, Ancash, Huari, Huascarán National Park, 1 km below Manto Mina, 3 km from Catac-Chavin road, 4300 m, 4 July 1985, *D.N. Smith & M. Buddensiek 11023* (MO331445!); Ancash, Huari, Huascarán National Park, 1 km below Manto Mina, ca. 3 km from Catac-Chavín road, 4300 m, 4 July 1985, *D.N. Smith & M. Buddensiek 11023* (USM69878!); Huánuco, Huacaybamba, Huacaybamba, Cerro Saquicocha Punta, 4000 m, 15 May 2017, *D.B. Montesinos 5608* (B100761319!); Huanuco, Huamalies, Llata, Waita-Waita, 4400 m, 9 May 2017, *D.B. Montesinos 5489* (B100761262!); Junin, La Oroya, Pachacayo, 4200 m, 7 April 1974, *P. Gutte 2267b* (LPZ!); Junin, La Oroya, Pachacayo, Haciendo Piñascochas, 4450 m, 3 April 1974, *P. Gutte 3228b* (LPZ!).

**11. *Paronychia ellenbergii*** Chaudhri, Revis. Paronychiinae 179–180, 1968. Figure 2E.

Holotype: Peru, Cuzco, Collutaro, Anden-Hochland, Südperus, Regenzeit. 30–35° SW Collutaro, OSO v. Cusco. 3400 m. 14 April 1957, *Ellenberg 1033* (U0008329, https://plants.jstor.org/stable/viewer/10.5555/al.ap.specimen.u0008329, accessed on 19 February 2023).

*Specimens examined*: PERU, Cusco, Saqsayhuaman, 3580 m, 1974, *P. & G. Gutte 1825b* (LPZ!); Huanuco, Huanuco, Huallintusha, 2400 m, 21 April 1974, *P. & G. Gutte 2772b* (LPZ!); Lima, Canta, Río Chillón, 2400 m, 8 January 1972, *G. Müller* et al. *1176b* (LPZ!).

**12. *Paronychia franciscana*** Eastw. Bull. Torrey Bot. Club 28(5): 288–289, 1901.

Lectotype (designated by Chaudhri [11]): U.S.A., California, San Francisco, Presidio, 22 April 1887, *Greene s.n.* (UC120270, *non vidi fide* Chaudhri [11] (page 153)); isolectotypes: US589370! (http://n2t.net/ark:/65665/3b1b190cd-30dc-42fc-bd23-f922d212a9d0, accessed on 19 February 2023), NY00342574 (https://plants.jstor.org/stable/viewer/10.5555/al.ap.specimen.ny00342574?loggedin=true, accessed on 19 February 2023), GH01714785! (https://s3.amazonaws.com/huhspecimenimages/JPG-Preview/01714785.jpg, accessed on 19 February 2023).

*Note.* Chaudhri [11] stated “Type: California: San Francisco, Presidio, 22.iv.1887, E. L. Greene (Lectotype UC!)”. According to Art. 7.11 of ICN, although the phrase “designated here” was not reported, the typification is formally correct.

On the basis of the protologue by Eastwood [38], *locus classicus* is “…grassy hillrocks at the Presidio, San Francisco” (the author also indicated syntypes (Art. 9.6 of ICN) by stating “there are specimens collected in the herbarium of the California Academy of Science, collected by Dr. E. L. Greene, Mrs. Brandegee, and Miss Evelina Cannon; also, one from Bodega Port collected by the writer”). Note however that, according to [39], *Paronychia franciscana* is native to Chile, and it is alien in California.

*Specimens examined*: Herbarium Musei Chilensis, December 1861, *s.n.* (illegible label, LP050837!); *iibidem* (LP036403!).

**13. *Paronychia glabra*** Montesinos, E.Rodr. & Iamonico, *sp. nov.*

Holotype: PERU. Amazonas department, Chachapoyas province, Leimebamba district, locality Zona 1, La Boveda, Tajopampa; depresiones de origen kárstico, dense forest, soil with deep layers of organic matter, 3672 m, 18 June 2018, *D.B. Montesinos 6936* (holotype HUT61788!, isotypes: B101098617!, HSP!, RO!). Figure 3A–F.

*Diagnosis. Paronychia glabra* differs from *P. hieronymi* [40] by the larger size of the stems and internodes, the entire and glabrous leaves, aristulate leaf apex, upper stipules bearing trichomes, long and densely scarious sepals, longer petals and anthers, shorter style size, and by the larger ovary size (Figure 4).

*Description*. A perennial herb with a thick woody caudex; stems infrequently prostrate or more commonly pendulous, pendent, or suspended, more or less clothed with older leaves down to the base, 20–100 cm long, decongested, spreading from the base and much branched over their whole length, bark yellowish; internodes 8–10 mm long, stout, densely pubescent and glabrous, mostly distant from the leaves, with the stipules often tapering, branches leafless in the basal part, and usually greenish–yellowish. Leaves shortly petioled, ovate to lanceolate, stiff, 7–9 × 2–3 mm, coriaceous, pale green turning bright yellow to brown with age (older leaves persistent), glabrous, densely covered by minuscule white-translucid glands on the underside and margins, with less frequency on the upperside, apex aristate, bearing 0.9–1.1 mm long stout mucro, dark green turning reddish towards the tip, with the margin plain, not tapering, margins entire, rarely shortly ciliate, petioles 0.1–0.3 mm long; stipules ovate to broadly lanceolate, apex acute to attenuate, becoming bifid, 5–7 × 1.5–2.5 mm, shorter than the leaves, glabrous and with entire margins except at the terminal sections bending and overlapping the flowers where the margins are usually hirsute, lamina pale white to translucid white; bracts ovate, usually bending the flowers, with entire margin becoming very narrow, 1.8–2.2 × 1.4–1.7 mm; glomerules in axial fascicules, 1–3 per axis, not congested, subterminal; flowers subsessile or more frequently sessile; sepals broadly ovate, densely hirsute towards the base and apex, with entire margins, greenish-red colored, 2–2.8 × 1–1.5 mm, awns acuminate, glandular and glabrous; petals reduced, ovate, 1.2–1.8 × 0.9–1.1; filaments 0.3–0.6 mm long; stamens 10; anthers 0.5–0.7 × 0.15 mm, oblong and yellowish-white; ovary ovoid, 0.7–0.9 mm, with a long style, 0.3–0.6 mm. Fruits ovate, glabrous, less than 1 mm in size; seeds suborbicular, shiny.

*Etymology*. The specific epithet *glabra* refers to the lustrous and hairless leaves of the plant, which is a unique feature almost unknown among South American species of *Paronychia*.

*Taxonomic discussion. Paronychia glabra* is unique among the known neotropical species because of the long stems bearing glabrous and entire leaves. A characteristic feature that is well known in most of the South American species is the leaf bearing trichomes and ciliate margins. *P. glabra* differs from its closest relative *P. hieronymi* by Pax [40] by (Figure 4 and Figure 5) the larger size of the stems (100 cm in *P. glabra* vs. 10 cm in *P. hieronymi*), leaves in *P. glabra* being glabrous and bearing entire margins which can rarely be shortly hirsute (vs. ciliated margins in *P. hieronymi*), aristulate leaf apex (*vs*. acute-acuminate in *P. hieronymi*), upper stipules bearing trichomes (*vs.* entire stipules), glomerules bearing 1–3 flowers (*vs*. 3–13 flowers in *P. hieronymi*), larger size of the flowers (2.2–3.0 mm vs. 1.75–2.00 mm in *P. hieronymi*), by the longer and densely scarious sepals (2.0–2.8 mm vs. 1.0–1.5 mm and semi-scarious in *P. hieronymi*), longer petals (1.2–1.8 mm vs. 0.75–0.90 mm in *P. hieronymi*), longer anther (0.5–0.7 mm vs. 0.35 mm in *P. hieronymi*), longer style size (0.3–0.6 mm vs. 0.6–0.9 mm in *P. hieronymi*), and by the larger ovary size (0.7–0.9 mm vs. 0.5–0.6 mm long in *P. hieronymi*). Moreover, the species differ in their ecology and distribution, in which *P. glabra* develops on karstic soils with heavy rainfall in N Peru (latitude 6° S) and *P. hieronymi* in NW Argentina, apparently in the Selva de las Yungas ecosystem located at latitude 24° S, which is only known from the type locality region, and possibly nearby geographic areas according to the scarce material found.

Another similar species, that occurs within the same geographic areas of *Paronychia gabra*, is *P. macbridei*. *P. glabra* differs by (Figure 5 and Figure 6) the larger plant size (*vs*. stems of about 20 cm long in *P. macbridei*), glomerules bearing 1–3 flowers in *P. glabra* vs. 3–5 flowers in *P. macbridei*, shorter internodes (8–12 mm long vs. 17 mm long in *P. macbridei*), larger leaf size (7–9 mm vs. 3.0–6.5 mm long in *P. macbridei*), leaf lamina glabrous vs. pubescent in *P. macbridei*, larger petal size (1.2–1.8 mm vs. 0.55–0.6 mm in *P. macbridei*), and by the larger style size (0.3–0.6 mm vs. 0.25–0.3 mm in *P. macbridei*). *P. glabra* is less similar to *P. communis* Cambess. by the smaller leaf size and texture (7–9 mm in *P. glabra* vs. 15 mm), larger stipule size (5–7 mm long vs. 5 mm in *P. communis*), and larger sepal size (2–2.8 × 1–1.5 mm vs. 1–1.2 × 0.45 mm in *P. communis*).

*Distribution and ecology. Paronychia glabra* is found in the mountain regions of north Peru, in what is denominated as “jalca” by the abundancy of tussock grasses with forest patches which receive a relatively considerable amount of rain that surpasses 1000 mm. The species is distributed at an altitude of 3690 m and is believed to be in the range of 3600–3750 m. Several introduced species have been observed in community with natives and endemics in the area according to [41], moreover, the geographical region where the species develops is prone to intense slope burning for conversion to agriculture, this is permanently degrading the terrains, causing erosion and loss of diversity. The type material (*Montesinos 6936*) was collected from rocky slopes with elements of karst origin, environments that are also found in south-east Cajamarca [42], which consist of quartzite and sandstones which are widely distributed in tropical areas of South America, being subject to dissolution processes, generating significant caves and karst [43].

*Paratypes.* PERU, Amazonas department, Chachapoyas province, Leimebamba district, Rocky slopes with scrubs at Mesapampa-Tajopampa, 3696 m, 6°50′18″ S, 77°49′11″ W, 14 August 2016, *D.B. Montesinos & L. García-Llatas 4980* (HSP9574!, HUT60205!, OV227247!); Amazonas department, Chachapoyas province, Leimebamba district, Cima montaña al sur de la Boveda, Pajonales de Jalca recientemente incinerados, 3691 m, 6°50′18″ S, 77°49′23″ W, 22 June 2018, *D.B. Montesinos 7109* (HSP!, B101098619!).

**14. *Paronychia hartwegiana*** Rohrb. Linnaea 37: 204, 1871–1873 [1872]. Figure 7A.

Lectotype (designated by Chaudhri [11] as “neotype”, here corrected according to Art. 9.10 of ICN; second-step lectotypification (Art. 9.17) here designated): Ecuador, ad ripas fluminis prope urbem Loxa, November 1842, *Hartweg 829* (G201985/2!, image available at http://www.ville-ge.ch/musinfo/bd/cjb/chg/adetail.php?id=200942&base=img&lang=en, accessed on 19 February 2023); isolectotypes: BR0000006981488! (https://plants.jstor.org/stable/viewer/10.5555/al.ap.specimen.br0000006981488, accessed on 19 February 2023), BM000803895! (https://plants.jstor.org/stable/viewer/10.5555/al.ap.specimen.bm000803895, accessed on 19 February 2023), F30176! (https://fm-digital-assets.fieldmuseum.org/360/904/30176.jpg, accessed on 19 February 2023), FI011232! (http://parlatore.msn.unifi.it/types/search.php, accessed on 19 February 2023), P00335807! (https://plants.jstor.org/stable/viewer/10.5555/al.ap.specimen.p00335807, accessed on 19 February 2023).

*Notes.* Chaudhri [11], by stating that the holotype was destroyed (“holo. B destroyed”), proposed to neotypify the name *Paronychia hartwegiana* on a specimen deposited at G (“S. Equador: ad ripas fluminis prope urbem Loxa, -/1842, *Hartweg* 829”), also reporting an isoneotype at FI. By checking the G herbarium, we traced two *Hartweg* specimens no. 829 (G201985/1 and G201985/2) which are are both part of the original material for this name. In addition, the FI specimen (FI011232) is part of the original material. So, these three specimens are eligible as lectotypes, not neotypes as stated by [11]. According to Art. 9.10 of ICN, Chaudhri’s uses of “neo-” (=neotype) and “iso-” (=isoneotype) are errors and they should be corrected to lecto- and isolectotype. Furthermore, since two G specimens were found, but [11] did not specify any barcode, his typification is to be considered as a first step and we here propose the second-step typification (see Art. 9.17 of ICN) using G201985/2 which bears a plant with more flowers and leaves. Note that the other G specimen (G201985/1) bears a plant collected on a different date than G201985/1 (“*1843*” according to the original label in the bottom-right corner of the sheet) and, therefore, it cannot be considered as a duplicate of the lectotype. We also traced further isolectotypes at BR (BR0000006981488), BM (BM000803895), F (F870647!), and P (P00335807). The types match the diagnosis and correspond to the current concept of the species (see e.g., [11]).

*Specimens examined*: ECUADOR, Azuay, Cuenca, Loja road, N slopes of Valle del Río León, 2700 m, 18 January 1983, *B. Lojtnant 15117* (MO6047068!); Loja, vicinity of Loja, 29 September to 3 October 1918, *J.N. Rose* et al. *23262* (NY!); PERU, Amazonas, Luya, Tingo, Path from Kuelap to Tingo, 2057 m, 12 August 2016, *D.B. Montesinos & L. García-Llatas 4957b* (B101156706!); Amazonas, Luya, Tingo, 1850 m, 29 August 1974, *P. Gutte & López 3739b* (LPZ!).

**15. *Paronychia hieronymi*** Pax, Bot. Jahrb. Syst.18: 34. 1894 ≡ *Paronychia chilensis* DC. subsp. *chilensis* var. *hieronymi* (Pax) Hosseus in Bol. Acad. Nac. Cien. Cordoba 26:84 (1921).

Lectotype (here designated): Argentina, Tucuman, La Ciénaga, 10–14 January 1874, *Lorentz & Hieronymus 581* (GOET!), Figure 8.

*Notes.* Pax [40] validly proposed this species providing a detailed description; moreover, the following syntype was indicated (Art. 9.6 of ICN): “Tucuman, La Ciénaga (Hieronymus et Lorentz n. 581a. ―10. ―17. Jan. 1874”. Chaudhri [11] stated “Type: Argentine, locality unspecified, Hieronymus in iii.1873, (holo. B, destroyed)” and, as a consequence, proposed to neotypify the name *Paronychia hieronymi* using a specimen preserved at G. However, based on this statement, it is clear that Chaudhri did not check the protologue and, probably, he did not check GOET. However, we traced at GOET (where Lorentz’s collection is preserved; [44]) one specimen collected at “*La Ciénaga, … Tucuman, 10-14/1 1874*” by P. G. Lorentz and G. Hieronymus, as reported on the original label. This specimen is clearly part of the original material and eligible as a lectotype. According to Art. 9.19 of ICN, “The author who first designates … a neotype … must be followed, but that choice is superseded if *(a)* … in the case of a neotype, any of the original material is found to exist …”. So, Chaudhri’s neotype designation is here superseded and the found GOET specimen is here designated as the lectotype of the name *Paronychia hieronymi*. The type matches the diagnosis and corresponds to the current concept of the species (see e.g., [11]).

*Specimens examined*: ARGENTINA, Jujuy, Tres Cruces, 2 February 1944, *A. Soriano 633* (LP909490!); Tucumán, Tafí, Cumbres Calchaquíes, Peñas Azules, 19 February 1974, *E.M. Zardini 335* (LP!); Salta, *Grisebach 11/77* (K!).

**16. *Paronychia johnstonii*** Chaudhri, Revis. Paronychiinae: 169–170. 1968.

Holotype: Northern Chile, Prov. Antofagasta, Depart. Taltal, Aguada de Cachinalcito, 28 November 1925, *I. M. Johnston 5184* (G, *not see fide* [11]; isotypes: S-R4024! (https://plants.jstor.org/stable/viewer/10.5555/al.ap.specimen.s-r-4024, accessed on 19 February 2023), US00589353, (https://plants.jstor.org/stable/viewer/10.5555/al.ap.specimen.us00589353, accessed on 19 February 2023), GH00037822, (https://plants.jstor.org/stable/viewer/10.5555/al.ap.specimen.gh00037822, accessed on 19 February 2023), F0053313F, (https://plants.jstor.org/stable/viewer/10.5555/al.ap.specimen.f0053313f, accessed on 19 February 2023)).

= *Paronychia johnstonii* Chaudhri subsp. *johnstonii* var. *scabrida* Chaudhri, Revis. Paronychiinae: 170. 1968, *syn. nov.*

Holotype: Northern Chile, Prov. Antofagasta, Pept. Taltal: Vicinity of Aguada de Miguel Díaz, l–4 December 1925, *I.M. Johnston 5356* (GH00037823, https://plants.jstor.org/stable/viewer/10.5555/al.ap.specimen.gh00037823, accessed on 19 February 2023).

*Notes.* Chaudhri [11] described *Paronychia johnstonii* subsp. *johnstonii* var. *scabrida* as plants with smaller leaves, and leaves and sepals pubescent. However, this variety was described from the same Chilean province and we did not see any difference between the two taxa. So, we here propose to synonymize the two names.

**17. *Paronychia jujuyensis*** (Chaudhri) Iamonico & Montesinos, *comb. et stat. nov.* ≡ *Paronychia hieronymi* subsp. *hieronymi* var. *jujuyensis* Chaudhri. Meded. Bot. Mus. Herb. Rijks Univ. Utrecht 285: 187–188, 1968.

Holotype: Argentina, Prov. Jujuy, Loma del Tambo, Volcán, 2500 m, 22 February 1924, *Schreiter 2715* (LIL000450, https://plants.jstor.org/stable/viewer/10.5555/al.ap.specimen.lil000450, accessed on 19 February 2023).

*Notes. Paronychia jujuyensis* differs from *P. hieronymi* by the leaves being mostly acuminate, pungently mucronate, and longer mucro (vs. Mostly acute, not pungent in *P. hieronymi*), longer stipule (6 mm vs. 4 mm long in *P. hieronymi*), and longer flowers ((2.50–)2.70–3.25 mm long vs. (1.75–)2.00–2.40 mm long in *P. hieronymi*) (Figure 5 and Figure 9). *P*. *jujuyensis* is an endemic taxon with restricted distribution area, known from *locus classicus* only (Loma de Tambo, Jujuy, NW Argentina). We here propose to treat it as a separate species [11].

**18. *Paronychia libertadiana*** Chaudhri, Revis. Paronychiinae 182–183, 1968. Figure 2F.

Holotype: Peru, La Libertad, Prov. Huamachuco, Río Marañón canyon, summit above Aricapampa, road to Huamachuco, 3970 m, 10 August 1964, *P.C. Hutchinson et al. 6269* (F0042718F!, image available at https://plants.jstor.org/stable/viewer/10.5555/al.ap.specimen.f0042718f, accessed on 19 February 2023); isotypes: NY00277919 (https://plants.jstor.org/stable/viewer/10.5555/al.ap.specimen.ny00277919, accessed on 19 February 2023), USM000300 (https://plants.jstor.org/stable/viewer/10.5555/al.ap.specimen.usm000300, accessed on 19 February 2023).

*Specimens examined*: PERU, La Libertad, Sánchez Carrión, Laguna Negra, al pie del Nevado de Huaylillas, 4200 m, 22 May 2001, *A. Sagastegui & M. Zapata 16530* (MO!); La Libertad, Otuzco, Cerro Ragache (Salpo), 3200 m, 23 May 1984, *A. Sagástegui* et al. *11605* (MO3244412!); Ancash, Huaylas, Huascarán National Park, Quebrada Alpamayo, 4350–4250 m, 9 March 1985, *D.N. Smith* et al. *9818* (MO3214451); Cajamarca, Cajabamba, Cajabamba, Luchubamba, 3800 m, 17 November 1983, *A. Sagástegui* et al. *11164* (MO3148028!); La Libertad, Sánchez Carrión, Desvío a Huaguil, Chugay Molino Viejo, 3800 m, 4 May 2003, *A. Sagástegui* et al. *17205* (HUT40944!); La Libertad, Santiago de Chuco, Quiruvilca, parte alta del río Chicama, al O del cerro Yanahuanca, 3976 m, 29 April 2003, *A. Cano* et al. *13000* (USM!); Huánuco, Lauricocha, San Miguel de Cauri, 3865 m, 8 November 2002, *F. Salvador* et al. *537* (USM!); Ancash, Huaylas, Huascarán National Park, Auquispuquio area of ruins, 3800–3900 m, 7 April 1986, *D.N. Smith* et al. *12001* (USM102451!); Ancash, Carhuaz, Pampa de Ulta, 3600–3650 m, 14 April 2001, *A. Cano* et al. *11170* (USM!); La Libertad, Cajamarca, Cajabamba, Cajabamba, Luchubamba, 3800 m, 17 November 1983, *A. Sagástegui* et al. *11164* (F1982156!); La Libertad, Bolivar, Uchumarca, Chivane, 3500–3600 m, 9 July 2010, *Bussmann* et al. *16781* (MO6589787!, http://legacy.tropicos.org/Image/100369360, accessed on 19 February 2023); Ancash, Huari, Huacchis, Tocana, Cerro Cuncush, 3900 m, 13 May 2017, *D.B. Montesinos 5569* (B100745228!); La Libertad, Bolivar, Uchumarca, Collpacucho, 3850 m, 7 November 2013, *Bussmann* et al. *17900* (MO6630236!, http://legacy.tropicos.org/Image/100415746, accessed on 19 February 2023); La Libertad, Bolivar, Uchumarca, Ucuncha, Cerro Fila de Andonsa, 3700–3800 m, 12 November 2013, *Bussmann* et al. *18191* (MO6630247, http://legacy.tropicos.org/Image/100415729, accessed on 19 February 2023); Amazonas, Leymebamba, Atalaya, along path to La Esperanza, 3450–3550 m, 28 June 2010, *A. Glenn* et al. *398* (MO6630237!, http://legacy.tropicos.org/Image/100415694, accessed on 19 February 2023).

**19. *Paronychia limaei*** Chaudhri, Revis. Paronychiinae: 189–190. 1968.

Holotype: Peru, Lima, Huarochiri, Infiernillo, 3250 m, 23 April 1939, *T.H. Goodspeed* et al. *11510* (UC657967, image available at https://plants.jstor.org/stable/viewer/10.5555/al.ap.specimen.uc657967, accessed on 19 February 2023); isotype: GH00037824 (https://plants.jstor.org/stable/viewer/10.5555/al.ap.specimen.gh000378249, accessed on 19 February 2023.

*Specimens examined*: PERU, Lima, Yauyos, Laraos, 26 February 1991, *H. Beltrán 215* (USM140294!).

**20. *Paronychia macbridei*** Chaudhri, Revis. Paronychiinae 183–184. 1968. Figure 7B.

Holotype: Peru: Huanuco, 2300 m, 5–8 April 1923, *J. F. Macbride 3243* (F0042719F!, image available at https://plants.jstor.org/stable/viewer/10.5555/al.ap.specimen.f0042719f, accessed on 19 February 2023), isotype: G00227052 (https://plants.jstor.org/stable/viewer/10.5555/al.ap.specimen.g00227052, accessed on 19 February 2023).

*Specimens examined*: PERU, Amazonas, Bongará, 2 km below Campamento Ingenio, along río Utcubamba, 1250–1275 m, 28 January 1964, *P.C. Hutchinson & J.K. Wright 3843* (MO1831128!); Cajamarca, Contumazá, Contumazá-Cascabamba, 2700 m, 12 April 1981, *Sagasteguí* et al. *9976* (MO3100306); Peru, Cusco, Calca, Huambutio, San Salvador, 3200 m, June 1986, *R. Dueñas 63* (MO3910968!); La Libertad, Santiago de Chuco, alrededores de Santiago de Chuco, 2800 m, 13 June 1984, *A. Sagástegui 11784* (MO5198504!); Huánuco, Yarowilca, Choras, Restos Arqueológicos de Garu, 3756 m, 7 May 2017, *D.B. Montesinos 5450* (B100761271!); Junín, Tarma, 27 January 1968, *W. Schwabe 68532* (B101156360!); Huánuco, Yarowilca, Aparicio Pomares, Sahuay Archaeological Site, 3666 m, 29 July 2016, *D.B. Montesinos 4862* (B100745268!); Ancash, Huari, San Pedro de Chana, Abra que divide dos departamentos, 4053 m, 15 May 2018, *D.B. Montesinos & G. Sancho 6379* (B101098589!); Huancavelica, Pampas, 3200 m, 7 August 1973, *P. Gutte 1059d* (LPZ!); Lima, Canta, Culluhuay, 4000 m, 21 June 1974, *P. Gutte 3210b* (LPZ!); Cajamarca, Cajamarca, Hualgayoc, km38, 4000 m, 9 January 1979, *G. Mülle & P. Gutte 8991* (LPZ!).

**21. *Paronychia mandoniana*** Rohrb., Linnaea 37: 208, 1871–1873.

Lectotype (here designated): Bolivia, Prov. Omasuyes, viciniis Achacache, cerro de Avichaca, in petrosis reg. alp., 4000 m, January 1861, *G. Mandon 994* (K000486391!, image image available at https://plants.jstor.org/stable/viewer/10.5555/al.ap.specimen.k000486391, accessed on 19 February 2023); isolectotypes G00227050 (https://plants.jstor.org/stable/viewer/10.5555/al.ap.specimen.g00227050, accessed on 19 February 2023), G00227051 (https://plants.jstor.org/stable/viewer/10.5555/al.ap.specimen.g00227051, accessed on 19 February 2023), P00335795 (https://plants.jstor.org/stable/viewer/10.5555/al.ap.specimen.p00335795, accessed on 19 February 2023).

*Notes.* Rohrbach [45], after a detailed description, listed the following syntypes (Art. 9.6 of ICN): “prope Potosi (d’Orbigny 1480!)”, “in provincia Omasuyos, viciniis Achacache, cerro de Avicacha, in petrosis, reg. alp. 4000 m. (Mandon 994!)”, and “in Peruviae montosis prope Azangaro (Lechler pl. peruv. 1760!)”. We traced these syntypes at G (*Mandon 994* (G00227050 and G00227051)), K (*Lechler 1760* (K000486392; image at https://plants.jstor.org/stable/viewer/10.5555/al.ap.specimen.k000486392, accessed on 19 February 2023) and *Mandon 994* (K000486391)), and P (*Mandon 994* (P00335795) and *d’Orbigny 148* (P00712731; image at https://plants.jstor.org/stable/viewer/10.5555/al.ap.specimen.p00712731, accessed on 19 February 2023)). Since [45] dedicated the species to G. Mandon, we prefer to select a specimen originally collected by him. Among G00227050, G00227051, K000486391, and P00335795, we here designate K000486391 as the lectotype of the name *Paronychia mandoniana* since it bears plants with many flowers whose features are important in the identification of the *Paronychia* species [11]. The types match the diagnosis and correspond to the current concept of the species (see e.g., [11]).

*Specimens examined*: BOLIVIA, Omasuyos, Cerro de Aruchaca, 4000 m, January 1861, *G. Mandon 994* (K!); Apolobamba-Kordillere, oberhalb des Ortes Charazani, 4160 m, 15 March 1983, *X. Menhofer X-2235* (LPZ!); PERU, Puno, Huancané, 3900 m, 1 March 1948, *P. Aguilar 243* (USM18893!); Tacna, Tarata, Cordillera del Barroso, 3800–4100 m, 26 March 1998, *M.I. La tore 2131* (USM159352!); Cusco, Urubamba, Chincheros, Summit of Antakillqa, 4500 m, 20 January 1982, *E.W. Davis* et al. *1718* (USM!); Junin, Atocsaico, 4000 m, 21 April 1982, *Tiller/Maas 324* (USM!).

**22. *Paronychia membranacea*** Muschl. Bot. Jahrb. Syst. 45(4): 460. 1911.

Lectotype (designated by Chaudhri [11] as ”isotype”, here corrected according to Art. 9.10 of ICN): Perú, Cajabambo, Acancho, Ocros, Chonta, 4400 m, 1906, *A. Weberbauer 2784* (G00227054!, image available at https://plants.jstor.org/stable/viewer/10.5555/al.ap.specimen.g00227054, accessed on 19 February 2023).

= *Paronychia andina* A.Gray in Bot. Unit. St. Exped. 1:128 (1854) subsp. *andina*.

*Notes.* Muschler [46] provided a detailed description and listed some syntypes (Art. 9.6 of ICN), i.e., *Weberbauer 2784* and *Weberbauer 2536*, both collected in Peru in 1903 and deposited in “Herb. Berol.” (=Herbarium Berolinensis) according to the protologue. Chaudhri [11] reported “Type: Peru: Oeros, Chonta, 4400 m, Weberbauer 2784 (holo. B. destroyed; iso. G!).”. We traced a specimen at B (B10-0243309; image at https://plants.jstor.org/stable/viewer/10.5555/al.ap.specimen.b%2010%200243309, accessed on 19 February 2023) bearing a label reporting “*Perú*|*leg. A. Weberbauer n. 2784*”, but these scripts do not seem to be original. A clear original label was found on a sheet at G (G00227054): “Dr. A. Weberbauer: Flora von Peru|n° *2784 Paronychia membranacea Muschl.*|Depart.: *Acancho* Prov.: *Cajabambo … Ocros … Chonta*|Höhe ü. d. Meer: *4400 m*|*1906*”. Although on the B specimen there is a recent label (by M. E. Timaná, year 2004) stating “TYPE OF: *Paronychia membranacea*” we think that it cannot be regarded as original material for Muschler’s name. So, G00227054 is here considered as the lectotype of *Paronychia membranacea* and, according to Art. 9.10 of ICN, [11]’s use of “iso.” (=isotype) is an error and it should be corrected. *Weberbauer 2784* was traced at B (B10-0243309) and it is here designated as the lectotype for *P. membranacea*; an isolectotype is deposited at G (G00227054). The types match the diagnosis and correspond to the current concept of the species (see e.g., [11]). Note that both B10-0243309 and G00227054 specimens bear labels by, respectively, M. Timaná (year 2004) and N. Chaudhri (year 1968) who identified Weberbauer’s plant as *Paronychia andina* subsp. *andina*. We agree with this identification since the material observed (B100243309) shares the morphological characters such as the leaves being elliptic-oblong to ovate, internodes of about 1–2 mm, stipules ovate to oblong, 2–3(–4) × 1.5–2.0(–5.0) mm, sepals oblong, 1.3–1.5(–2.0) × 0.5–0.7 mm.

**23. *Paronychia microphylla*** Phil. Anales del Museo Nacional de Chile. Segunda Sección—Botánica 8: 26. 1891.

Lectotype (designated here): Chile, Usmagama, prov. Tarapacá, March 1885, *C. Rahmer s.n.* (SGO000001972 https://plants.jstor.org/stable/viewer/10.5555/al.ap.specimen.sgo000001972, accessed on 19 February 2023).

*Notes.* Chaudhri [11] cited a specimen “Chile: Prov. Tarapaca: Dept. Tarapaca: Cord. Quebrada de Quipisca, Noasa, ca. 3500 m, E. Werderman 1061 (GH, U)”, but no indication about the type was reported. A lectotypification is necessary, since Philippi [46] did not cite any holotype. The protologue [47] reports, after the diagnosis, the provenance as “Habitat in Usmagama in Provincia Tarapacá”. Therefore, the GH and U specimens cited by [11], which were collected in a different locality, are not part of the original material and cannot be considered as lectotypes. We traced a specimen at SGO (SGO000001972) matching the protologue; it is here designated as the lectotype of the name *Paronychia microphylla*. The type matches the diagnosis and corresponds to the current concept of the species (see e.g., [11]).

*Specimens examined*: ARGENTINA, Salta, Rosario de Lerna, Potrero de Linares, 17 March 1958, *A.L. Cabrera & J.M. Marchionni 13139* (LP902791!).

**24. *Paronychia muschleri*** Chaudhri, Meded. Bot. Mus. Herb. Rijks Univ. Utrecht 285: 168, 1968, *nom. nov. pro Paronychia rigida* Muschl. *non* Moench (1794). Figure 7C.

Lectotype: not designated.

*Notes.* Muschler [46] validly published *Paronychia rigida* giving a detailed description; moreover, seven specimens, collected by A. Weberbauer from 1902–1903 in Peru (syntypes according to Art. 9.6 of ICN) were cited. Muschler’s name is a later homonym (illegitimate under Art. 53.1 of ICN) of the previously published (year 1794) *P. rigida* Moench (note that Moench’s *P. rigida* is in turn an illegitimate and superfluous name, since the valid *Illecebrum capitatum* L. was reported as a synonym; see Art. 52.2 of ICN). Chaudhri [11] correctly proposed a *nomen novum* for Muschler’s *P. rigida* (see also Art. 6.11 of ICN), dedicating it to Muschler. According to Art. 7.4 of ICN, Chaudhri’s replacement name *P. muschleri* is to be typified by the type of Muschler’s *P. rigida*.

According to [48], Weberbauer’s herbarium and types are mainly deposited at B (mostly destroyed after the Second World War), MOL, and USM. Unfortunately, we could not trace any specimen which is part of the original material. However, we prefer to avoid a neotypification since we received no reply from MOL, despite requests being sent. So, the typification of *Paronychia rigida* remains open.

*Specimens examined*: BOLIVIA, La Paz, Murillo, 4.8 km al noreste de la autopista por el camino subiendo al Valle del Río Kaluyo (margen norte de La Paz), 4100 m, 28 February 1987, *J.C. Solomon 16197* (MO04915129!); La Paz, Murillo, Canyon, 1 km N of Ovejuyo, 3700–3900 m, 4 April 1986, *J.C. Solomon 15233* (MO3621558!); La Paz, Murillo, 4 km below the toll booths at El Alto on the Autopista, 3850 m, 6 January 1985, *J.C. Solomon 13047* (MO3249510!); La Paz, 4100 m, May 1932, *O. Buchtien 249* (MO1157850!); Bolivia, La Paz, ca. 1 km NW of Ovejuyo, 3700–3800 m, 21 March 1982, *J.C. Solomon 7218* (MO3099497!); La Paz, Ingavi, vicinity of Hacienda Lacaya, ca 20 km NW of Tambillo, near Lago Titicaca, 3850–3900 m, 18 January 1984, *A. Gentry* et al. *44368* (MO3231359!); Chuquisaca, oropeza, Santuario de Chataquilla, hacia Punilla ca 12 km, 3303 m, 25 February 2007, *E. Cervantes 133* (MO6063094!); La Paz, Ingavi, cantón Jesús de Machaca, comunidad Titicani-Iacaca, a 20 km de Guaqui, 3920 m, 10 April 1989, *Villavicencio* et al. *693* (B101156448!); La Paz, Bautista Saavedra, Charazani, Chajaya, 23 March 1992, *P. Gutte G202* (LPZ!); La Paz, Bautista Saavedra, Charazani, Moyapampa, 29 March 1992, *P. Gutte G250* (LPZ!); La Paz, Bautista Saavedra, Charazani, Chajaya, 4200 m, 7 April 1992, *P. Gutte G340* (LPZ!); PERU, Puno, Amantaní, 3900 m, 16 March 1948, *P. Aguilar 252* (USM18895!); Ancash, Yungay, Huascarán National Park, Llanganuco sector, María Josefa trail between Chinancocha and Pucayacu, 3700–3850 m, 7 May 1985, *D.N. Smith 10528* (USM102454!); Tacna, Tarata, camino a Caro, margen derecho de río Chacavira, 3070–3480 m, 5 December 1997, *M.I. La Torre 1797* (USM159357!); Puno, 3900 m, 9 February 1948, *P. Aguilar 13* (USM18883!); Ancash, Yungay, Huascarán National Park, ruins at Auquispuquio, 3750–3900 m, 8 April 1986, *D.N. Smith* et al. *12042* (MO3336402!); Arequipa, Caylloma, vicinity of Chivay, 3600–3800 m, 21 April 2006, *H. van der Werff* et al. *20953* (MO6124817!); Cusco, Cusco, San Salvador, Pachatusan, 4034 m, 4 April 2017, *D.B. Montesinos* et al. *5312* (B100745209!); Junín, Acopalca, Ututupalla, 4100 m, 27 June 1960, *G.W.H. Kunkel 5858* (B100540087!); Junín, Huancayo, Puna, 4000 m, 12 December 1960, *G.W.H. Kunkel 5269* (B100540718!); Moquegua, General Sánchez Cerro, Ubinas, Tassa town, perturbed terrain near road, 3818 m, 13 February 2016, *D.B.Montesinos* et al. *4486* (B100761155!); Apurimac, Cotabambas, Haquira, Suitururi, rocky slopes with shrubs, 4220 m, 30 March 2017, *D.B. Montesinos & D. Cornejo 5242* (B100843041!); Arequipa, Caylloma, Chivay, zwischen Yanque und Caylloma, 3600 m, 21 March 1972, *G. Mueller* et al. *2081b* (LPZ!).

**25. *Paronychia peruviana*** Chaudhri, Mededeelingen van het Botanisch Museum en Herbarium van de Rijks Universiteit te Utrecht 285: 180–181, t. 6, f. 1–3. 1968.

Holotype: S. Peru. Checayani, NE of Azangaro, 3950 m, 26 March 1957, *H. Ellenberg 262* (U0008331!, image available at https://plants.jstor.org/stable/viewer/10.5555/al.ap.specimen.u0008331, accessed on 19 February 2023).

*Specimens examined*: PERU, Puno, Titicaca lake, 18 February 1903, *A.W. Hill 415* (K17/H2456!); Puno, Juliaca, 12,500 ft, 23 January 1937, *D. Stafford 451* (K16/H2456/67!); Puno, Carabaya, Macusani, Laguna Ccata, 4372 m, 6 April 2019, *D.B. Montesinos & K. Chicalla 7469* (MOQ!).

**26. *Paronychia revoluta*** C.E.Carneiro & Furlan, Novon 14(1): 33–35, f. 1. 2004.

Holotype: Brazil, Rio Grande do Sul, Cambará do Sul, 29 January 1983, *L.P. de Queiroz & L.S.S. Faria 470* (HUEFS (Figure 1 in [13]); isotypes: ALBC, HRBS).

**27. *Paronychia sanchez-vegae*** Montesinos & Kool, Phytotaxa 334(1): 44–45, figs. 2A–K, 3A–C, 2018. Figure 7D.

Holotype: Peru, Amazonas, Luya, Lamud, Cerro Mito, path to the archaeological site of Pueblo de los Muertos, 2504 m, 6°06′13″ S, 77°54′14″ W, 11 August 2016, *D.B. Montesinos & L. García 4924* (HSP!); isotypes: B100761822!, CPUN!, CUZ!, F!, HCSM!, HUSA!, HUT!, MOL!, O227305!).

*Specimens examined*: PERU, Amazonas, alrededores del Campo de Aviación de Chachapoyas, 2500 m, 13 April 1950, *R. Ferreyra 7149* (USM18322!, US2049303!); Amazonas, Chachapoyas, bajada Boca Negra, 6 August 1962, *J. Soukup 4911* (US2565581!); Amazonas, Chachapoyas, 24 January 1965, *J. Soukup 5276* (US2471593!); Ancash, Yungay, Huascarán National Park, Quebrada Yanapaccha, a lateral of Quebrada Ranincuray, 3700–3840 m, 19 April 1985, *D.N. Smith* et al. *10473* (F1983196!, USM68871!); Amazonas, Cerro de Fraijaco, Huari-Huari, 3300–3450 m, 7 July 1948, *F.W. Pennell 15876-A* (USM!); Amazonas, Luya, Lamud, Pueblo de los Muertos, 2497 m, 11 August 2016, *D.B. Montesinos & L. García 4936* (B100761521!, HSP!, O227304!); La Libertad, Huamachuco, Laguna Sausacocha, near Huamachuco, 3100–3150 m, 24 August 2004, *P. Sklenar & M. Zapata 8618* (PRC!).

**28. *Paronychia setigera*** (Gillies ex Hook. & Arn.) F. Herm, Repert. Spec. Nov. Regni Veg. 42: 224. 1937 ≡ *Herniaria setigera* Gillies ex Hook. & Arn., Bot. Misc. 3: 337. 1833.

Type (lectotype designated by Chaudhri [11] (pag. 175) as ”holotype”, here corrected according to Art. 9.10 of ICN): Argentina. Prov. San Luis, El Aguadita, near La Punta de San Luis, 1823–1828, *Gillies s.n.* (K000486390!, image available at https://plants.jstor.org/stable/viewer/10.5555/al.ap.specimen.k000486390?loggedin=true, accessed on 19 February 2023); isolectotypes: E00265082!, E00265083! (both the isolectotypes occur on a single sheet, which is available at: https://plants.jstor.org/stable/viewer/10.5555/al.ap.specimen.e00265083?loggedin=true, accessed on 19 February 2023).

– *Paronychia australis* Hook. & Arn., Bot. Misc. 3: 337. 1833, *nom. inval.* (Art. 36.1b of ICN).

*Notes.* Chaudhri [11] stated “Type (of *H. setigera*): Argentine: Prov. San Luis, El Aguadita, nr. La Punta de San Luis, *Gillies s.n.* (holo. K!)”. However, Hooker and Arnott (1833: 337) did not indicate for *Herniaria setigera* (Art. 9.1 of ICN), just stating “El Aguadita, near La Punta de San Luis, *Dr. Gillies*” (syntype according to Art. 9.6 of ICN). By considering that part of Gillies’ herbarium and types are preserved at K [49], according to Art. 9.10 of ICN, Chaudhri’s use of “holo” (=holotype) is an error to be corrected to lectotype. Two isolectotypes were traced at E (E00265082 and E00265083).

Chaudhri [11] recognized *Paronychia setigera* subsp. *setigera* and *P. setigera* subsp. *cordobensis* Chaudhri (new proposed subspecies) based on the shape of leaves, size of flowers, and length of awns of sepals; *P. setigera* subsp. *setigera* was further classified into two varieties, i.e., *P. setigera* subsp. *setigera* var. *setigera* and *P. setigera* subsp. *setigera* var. *longiseta* Chaudhri based on size of flowers and sepals; finally, two subvarieties were recognized for this latter variety, i.e., *P. setigera* subsp. *setigera* var. *longiseta* subvar. *setigera* and *P. setigera* subsp. *setigera* var. *longiseta* subvar. *subglabra* Chaudhri based on leaf hairiness. The high phenotypic variability of *P. setigera* is complicated and, on the basis on our observations, we prefer to avoid a taxonomic conclusion about these taxa. So, we provisionally accept Chaudhri’s classification, awaiting further investigations. For the same reason, we also accept *P. setigera* subsp. *argillicola* [50].

*Specimens examined*: ARGENTINA, Buenos Aires, Cerro Bachicha, Cantera Marinotti y Casado, 11 November 1952, *M.M. Job s.n.* (LP903552!); Buenos Aires, Partido de Tornquist, Sierra de la Ventana, Parque Provincial, 6 October 1939, *A.L. Cabrera 5286* (LP30043!); Salta, Departamento Capital, Finca Castellanos, Quebrada de Castellanos, 15 km al NW de Salta (Capital), 1600 m, 14 October 1995, *C.R. Volponi 1033* (LP2473!); San Luis, Chacabuco, ruta provincial 1, arroyo la Aguada, 12 December 1991, *Múlgura 1128* (LP!); Buenos Aires, Patagones, Isla del Jabalí, Rincón del Banco, 24 December 1981, *T.M. Pedersen 13204* (MO3770507!); Chubut, Isla Leónes, 10 January 1940, *R. Santesson 142* (K!); BOLIVIA, La Paz, Bautista Saavedra, Charazani, Chullinia, 3000 m, 20 April 1993, *P. Gutte & B. Herzog G401* (LPZ!); La Paz, Bautista Saavedra, Charazani, Carijana, 1700 m, 29 April 1993, *P. Gutte & B. Herzog G584* (LPZ!); La Paz, Bautista Saavedra, Charazani, Chullinia, 26 March 1992, *P. Gutte B153* (LPZ!); La Paz, Bautista Saavedra, Charazani, Sacanagon, 17 April 1972, *P. Gutte G56* (LPZ!).

**28a. *Paronychia setigera*** subsp. ***setigera***.

**28a1.** *Paronychia setigera* subsp. *setigera* var. *setigera*.

**28a2.***Paronychia setigera* subsp. *setigera* var. *longiseta* Chaudhri, Revis. Paronychiinae: 176. 1968.

Holotype: Argentina, Buenos Aires, Balcarce, Sierra del Volcan, E. of Balcarce, 3 November l946, *B. Sparre 276A* (S-R4025, https://plants.jstor.org/stable/viewer/10.5555/al.ap.specimen.s-r-4025, accessed on 19 February 2023).

**28a2-I. *Paronychia setigera*** subsp. ***setigera*** var. ***longiseta*** subvar. ***longiseta***.

**28a2-II. *Paronychia setigera*** subsp. ***setigera*** var. ***longiseta*** subvar. ***subglabra*** Chaudhri, Revis. Paronychiinae: 176. 1968.

Holotype: Argentina, Mendoza, Tunuyan, Las Heras, 22 May 1938, *Ruiz Leal 5248* (LIL000453, https://plants.jstor.org/stable/viewer/10.5555/al.ap.specimen.lil000453, accessed on 19 February 2023).

**28b. *Paronychia setigera*** subsp. ***argillicola*** Pedersen, Fl. Il. Entre Ríos 3: 262, 1987.

Holotype: Argentina, Entre Ríos, concepción del Uruguay, Arroyo Isletas, 15 December 1961, *A. Burkart & S. Crespo 22980* (SI001241, https://plants.jstor.org/stable/viewer/10.5555/al.ap.specimen.si001241, accessed on 19 February 2023).

**28c. *Paronychia setigera*** subsp. ***cordobensis*** Chaudhri, Revis. Paronychiinae: 176. 1968.

Holotype: Prov. Cordoba: Dep. Punilla, Mallin, ca. 900 m, 19 February 1951, *J. Gutierrez 309* (LIL!).

*Specimens examined:* ARGENTINA, Misiones, San Pedro, El Soberbio, 500 m, 12 July 1957, *J.E. Montes 27469* (LP1971531!).

**29. *Paronychia ubinensis*** Montesinos, Phytotaxa 124(1): 50–52, f. 1A–J, 2, 2013. Figure 7E.

Holotype: Peru, Moquegua Department, General Sánchez Cerro Province, Ubinas District, Punku near Tassa locality, 16°10′30″ S, 70°42′21″ W. 4030 m, 6 April 2012, *D.B. Montesinos 3698* (WAG!, isotype USM!).

*Specimens examined:* PERU, Moquegua, General Sánchez Cerro, Ubinas, Punku-Tassa, 4030 m, 6 April 2012, *D.B. Montesinos 3698* (USM!); Moquegua, General Sánchez Cerro, Yunga, Sura-Ccasullama, 4480 m, 7 April 2012, *D.B. Montesinos 3717* (HSP!, USM!, HUSA!); Moquegua, General Sánchez Cerro, Ubinas, Matazo, 4584 m, 24 March 2013, *D.B. Montesinos 4028* (CPUN!); Moquegua, General Sánchez Cerro, Yunga, Sura, 4530 m, 26 March 2013, *D.B. Montesinos 4049* (HSP!); Moquegua, General Sánchez Cerro, Ubinas, Entre Pirhuani y Rancho, 4592 m, 10 February 2014, *D.B. Montesinos 4146, 4150* (HUT!).

**30. *Paronychia weberbaueri*** Chaudhri, Revis. Paronychiinae 190. 1968, *nom. nov pro Paronychia polygonoides* Muschl. (1911) non Gurke (1899). Figure 7F.

Neotype (designated by Chaudhri [11]): Peru, Ancash, Cajatambo, Ocros, 3500 m, 1906, *A. Weberbauer* 2702 (G00227049!, image available at https://plants.jstor.org/stable/viewer/10.5555/al.ap.specimen.g00227049, accessed on 19 February 2023).

= *Paronychia polygonoides* Muschl. Botanische Jahrbücher für Systematik, Pflanzengeschichte und Pflanzengeographie 45(4): 459–460. 1911, *nom. illeg.* (homonym; Art. 53.1 of ICN) *non* Gürke (1899).

*Specimens examined:* PERU, Ancash, Recuay, near Conococha, 4000 m, 26 January 1985, *D.N. Smith* et al. *9363* (USM!); Ancash, Huaylas, road from San Toribio to pass to Santa, 4665 m, 12 October 2007, *M. Weigend & H.H. Hilger 8904* (USM225118!); Ancash, Carhuaz, Huascarán National Park, quebrada Ishinca, 4370 m, 11 February 1985, *D.N. Smith* et al. *9438* (USM67810!, HUT021334!); Ancash, Recuay, Conococha, 4100 m, 3 November 1984, *A. Sagástegui* et al. *12345* (MO3328914!, HUT19429!); Cusco, Anta, Cillapuyu, El Chaccan, 3685 m, 13 April 1973, *G.R. Brunel 782* (MO!); Ancash, Recuay, Huascarán National Park, Quebrada Quenua Ragra, 4700–4600 m, 10 May 1985, *D.N. Smith* et al. *10669* (MO3314454!, HUT023289!); Ancash, Recuay, Huascarán National Park, Quebrada Quenua Ragra, 4500–4600 m, 11 March 1986, *D.N. Smith* et al. *11726* (MO3357560!, HUT21259!); Ancash, Recuay, Huascarán National Park, Quebrada Quenua Ragra, 4700–4850 m, 11 March 1986, *D.N. Smith* et al. *11763* (MO3333942!); Junin, Tarma, 4000 m, 29 June 1954, *O. Tovar 2341* (USM18911!); Moquegua, General Sánchez Cerro, Ichuña, 600 m NW of Ichuña locality, 3749 m, 7 April 2009, *D.B. Montesinos 2488* (USM!); Ancash, Recuay, Huascarán National Park, Quebrada Queshque, lateral valley toward Rio Pachacoto, 4600–4500 m, 18 March 1986, *D.N. Smith* et al. *11839* (HUT021680!); La Libertad, Santiago de Chuco, Laguna el Toro, 4100 m, 14 December 1972, *A. Sagástegui 9430* (HUT15457!); La Libertad, Santiago de Chuco, Pampas de la Julia, 4000 m, 13 May 2003, *M. Zapata* et al. *17182* (HUT40922!); Ancash, Huari, Huacchis, Tocana, Cerro Cuncush, 4192 m, 13 May 2017, *D.B. Montesinos 5557* (B100745233!); Junin, La Oroya, Piñascochas, 27 km SW von Pachacayo, 4280 m, 22 March 1974, *P. Gutte 2173b* (LPZ!).


**Species of *Paronychia* to be excluded from South America.**


*Paronychia argyrocoma* (Michx.) Nutt., Gen. N. Amer. Pl. 1: 160. 1818 subsp. *argyrocoma* ≡ *Anychia argycoma* Michx., Fl. Botr. Amer. 1: 113–114. 1803.

*Note.* This species occurs exclusively in North America. We found three South American specimens at MO (MO491351, MO1101271, and MO1459562) which were identified as *Paronychia argyrocoma* by J. Ricketson. However, due to the morphological features (pulvinate perennial, leaves elliptic-oblong to ovate, internodes of about 1–2 mm, stipules ovate to oblong, 2–3(–4) by 1.5–2.0(–5.0) mm long, sepals oblong, 1.3–1.5(–2.0) × 0.5–0.7 mm), Ricketson’s identification is mistaken and MO exsiccatum actually refers to *Paronychia andina*.

## Figures and Tables

**Figure 1 plants-12-01064-f001:**
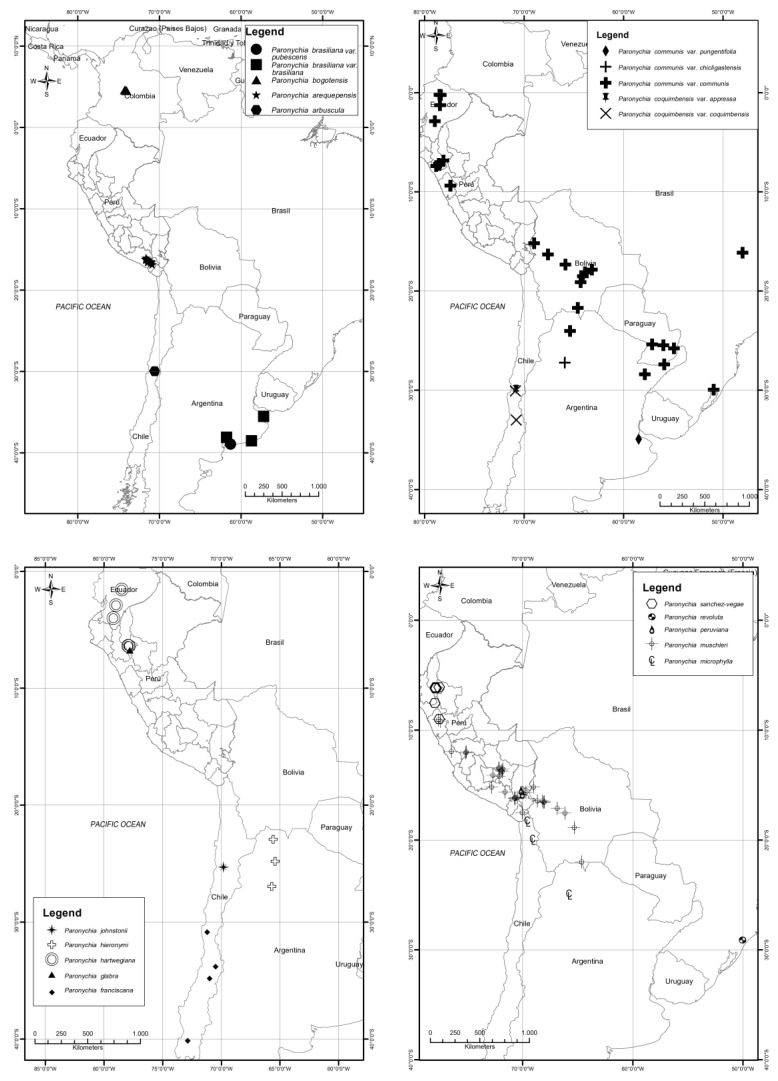
Distribution maps of *Paronychia* species occurring in South America.

**Figure 2 plants-12-01064-f002:**
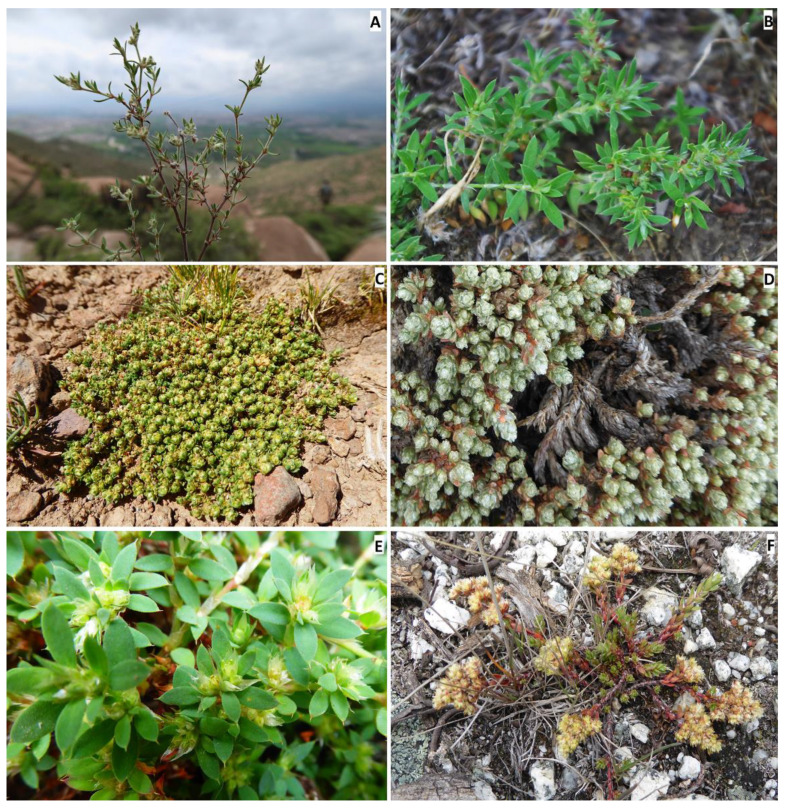
Habit images of different species of *Paronychia* occurring across the Andes. (**A**) Woody branch of the endemic *P. arequipensis* near El Batolito, Socabaya, Arequipa, 2300 m, (**B**) *P. chilensis* developing near Cerro La Campana, Valparaiso, Chile, 1000 m, (**C**) *P. compacta* subsp. *compacta* with characteristic mat-forming habit, near Tisco, m, Chivay, Colca Canyon, Caylloma, Arequipa, 4700 m, (**D**) *P. compacta* subsp. *purpurea* in Huascaran National Park, Ancash, Peru, 4600 m, (**E**) *P. ellenbergii* close-up, near Chincheros, Cusco, Peru, 3700 m, (**F**) *P. libertadiana* near Laguna Parón, in Huascaran National Park, Ancash, Peru, 4150 m. Photos: DBMT.

**Figure 3 plants-12-01064-f003:**
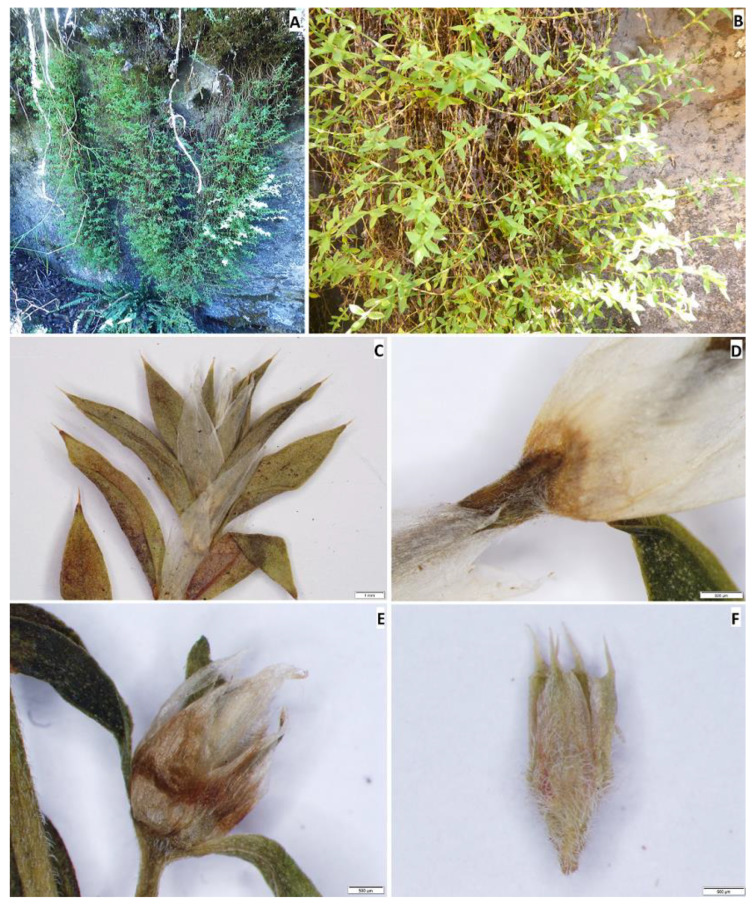
*Paronychia glabra*. (**A**) Habit of the species in rock crevices and karstic caves, Tajopampa, Chachapoyas, Amazonas, 3670 m, (**B**) branches showing the internodes, leaves, and stipules, (**C**) branch bearing glabrous leaves and notable stipules, (**D**) base of the node where the leaves and stipules are born, (**E**) stipules densely covering a single flower, (**F**) flower calyx. Photos: DBMT.

**Figure 4 plants-12-01064-f004:**
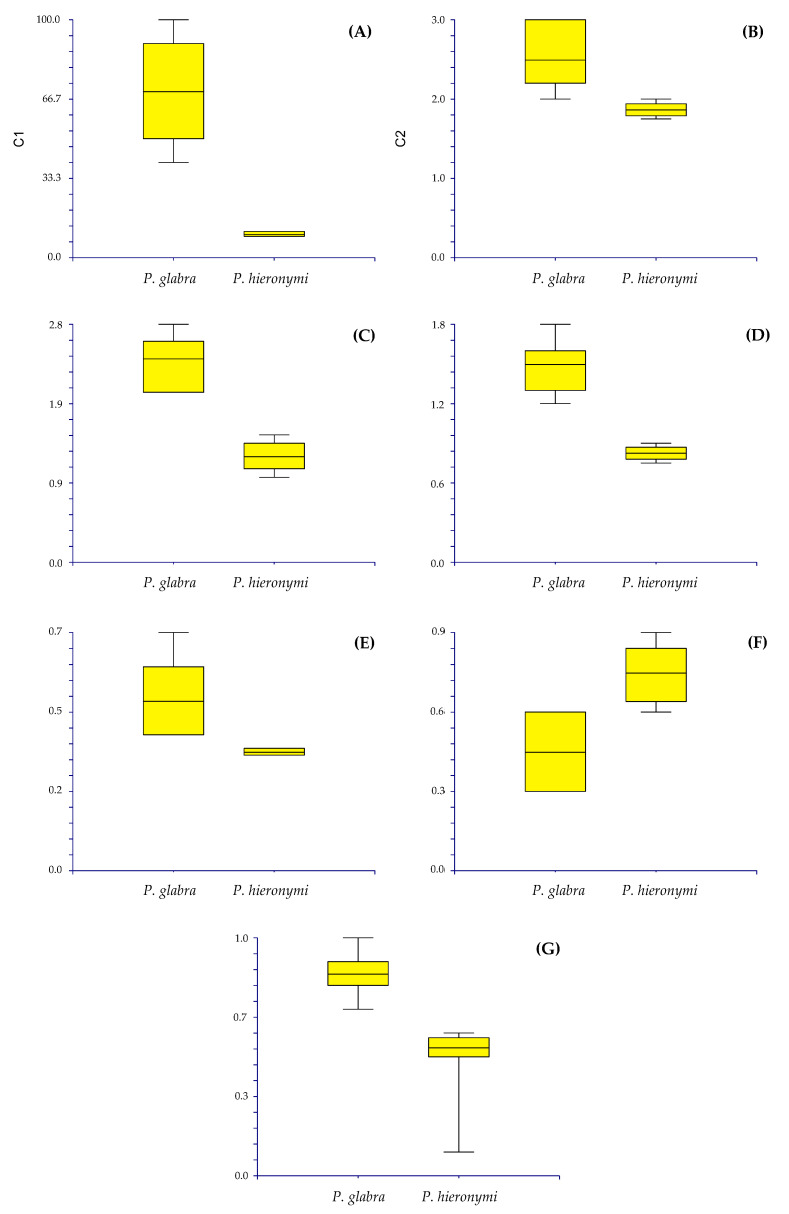
*Paronychia glabra* vs. *P. hieronymi*. Box plots illustrating the variability of: size of the stem (**A**), size of the flowers (**B**), length of the sepals (**C**), length of the petals (**D**), length of the anthers (**E**), length of the styles (**F**), length of the ovary (**G**). Yellow boxes illustrate interquartile ranges (25th–75th percentile) and medians (horizontal line); vertical lines are the whiskers (scores outside the middle 50%).

**Figure 5 plants-12-01064-f005:**
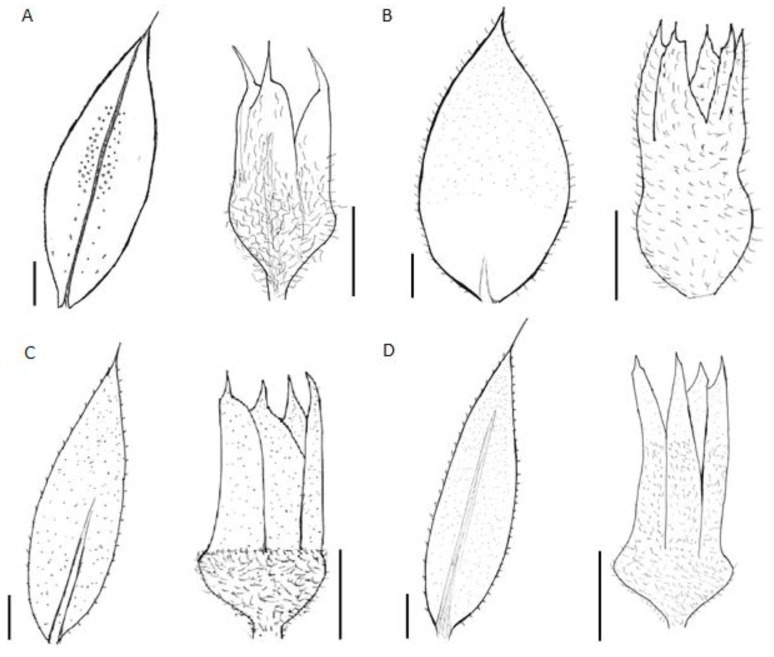
Comparison between leaves and sepals of *Paronychia glabra* (**A**), *P. hieronymi* (**C**), *P. macbridei* (**B**), and *P. jujuyensis* (**D**). Scale bars = 1 mm.

**Figure 6 plants-12-01064-f006:**
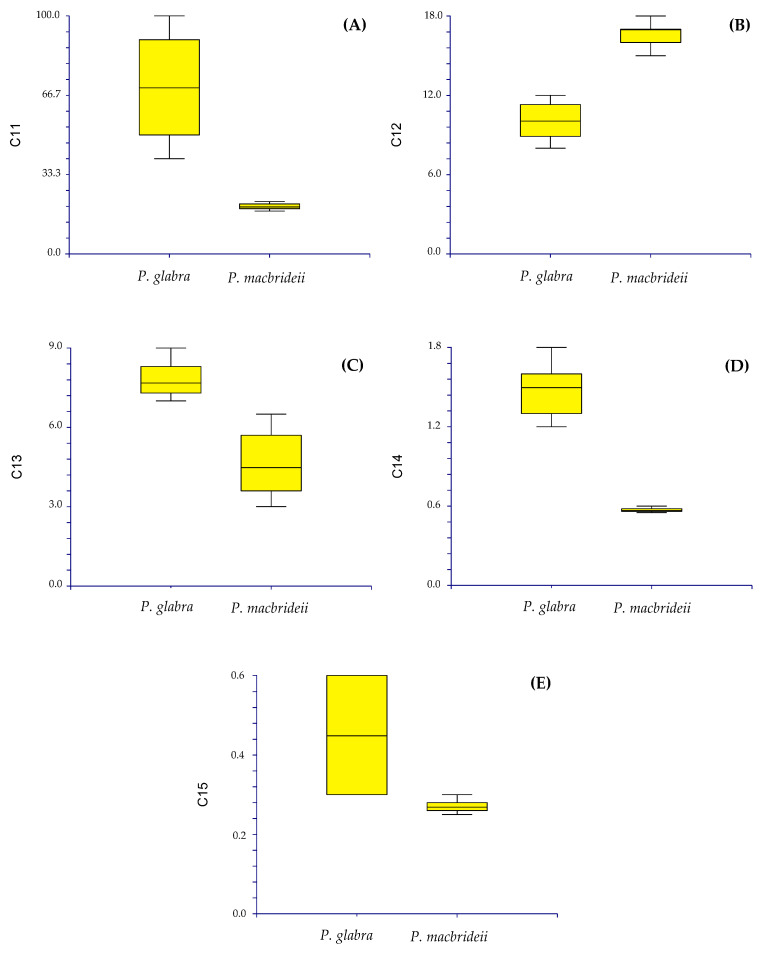
*Paronychia glabra* vs. *P. macbridei*. Box plots illustrating the variability of the diagnostic character: size of the stem (**A**), length of the internodes (**B**), length of the leaves (**C**), length of the petals (**D**), length of the styles (**E**). Yellow boxes illustrate interquartile ranges (=the range between the 25th and 75th percentile) and medians (horizontal line); vertical lines are the whiskers which represent the scores outside the middle 50% (i.e., the lower 25% of scores and the upper 25% of scores).

**Figure 7 plants-12-01064-f007:**
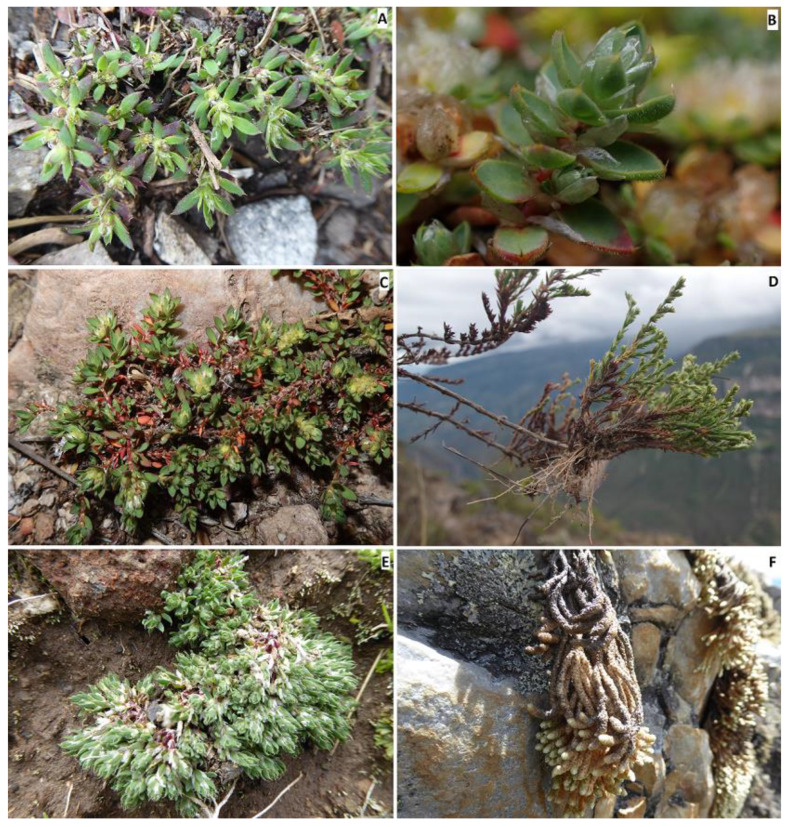
Photographs showing the different habit of *Paronychia* species occurring in South America. (**A**) *P. hartwegiana* near Laguna Cuicocha, Cotacachi, Imbabura, Ecuador, 3060 m, (**B**) close-up of *P. macbridei* growing on the Alto Marañon slopes, Huamalies, Huánuco, 3300 m, (**C**) *P. muschleri*, a relatively common species in the highlands of Moquegua, S Peru, 4000 m, (**D**) *P. sanchez-vegae* woody branch with roots, near Lamud, Chachapoyas, Amazonas, 2500 m, (**E**) *P. ubinensis* in the highlands of Yunga, Moquegua, 4400 m, (**F**) pendant branches of *P. weberbaueri* near Pastoruri, Huascaran National Park, Ancash, Peru, 4700 m. Photos: DBMT.

**Figure 8 plants-12-01064-f008:**
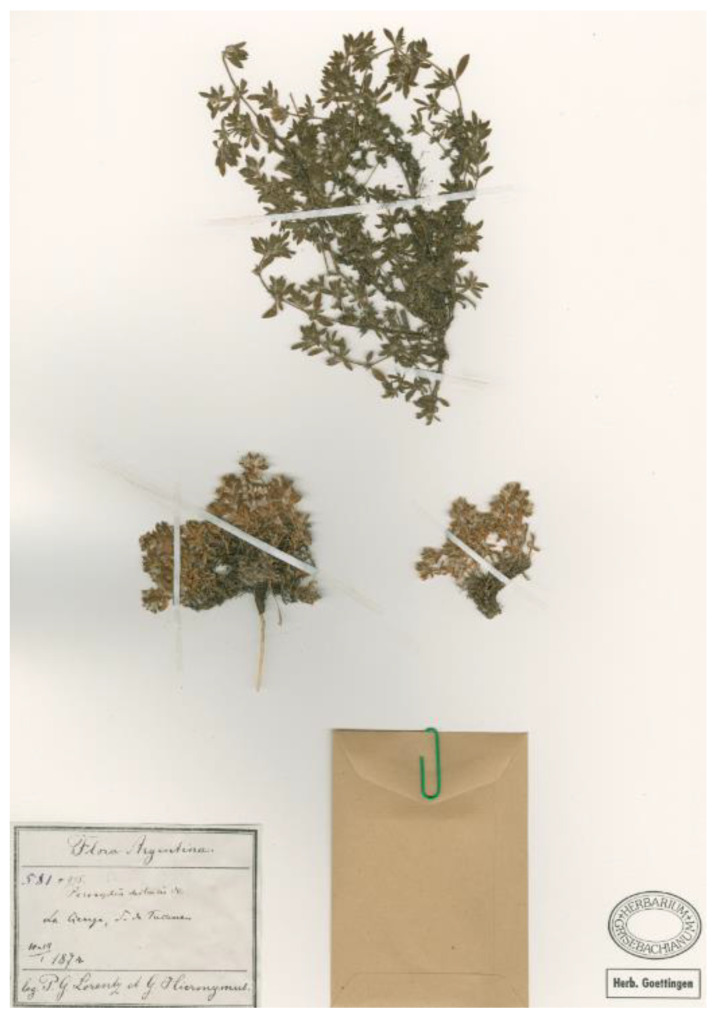
Lectotype of the name *Paronychia hieronymi* (GOET!).

**Figure 9 plants-12-01064-f009:**
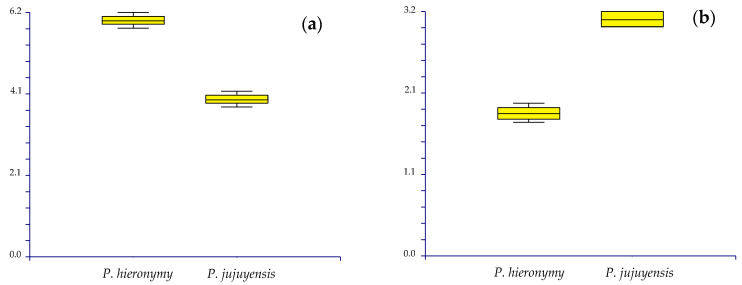
*Paronychia jujuyensis* vs. *P. hieronymi*. Box plots illustrating the variability of the diagnostic character: length of the flowers (**b**), length of the stipules (**a**). Yellow boxes illustrate interquartile ranges (=the range between the 25th and 75th percentile) and medians (horizontal line); vertical lines are the whiskers which represent the scores outside the middle 50% (i.e., the lower 25% of scores and the upper 25% of scores).

**Table 1 plants-12-01064-t001:** South American *Paronychia* taxa here recognized and their occurrence at national level. Abrbeviations of coutries names: Arg = Argentina; Bol = Bolivia; Bra = Brazil; Col = Colombia; Chi = Chile; Par = Paraguay; Per = Peru; Uru = Uruguay.

	Accepted Names	Arg	Bol	Bra	Col	Chi	Ecu	Par	Per	Uru
1	*Paronychia arbuscula*					x				
2	*Paronychia arequipensis*								x	
3	*Paronychia bogotensis*				x					
4	*Paronychia brasiliana* subsp. *brasiliana* var. *brasiliana*	x		x						x
	*Paronychia brasiliana* subsp. *brasiliana* var. *pubescens*	x								
5	*Paronychia cabrerae*	x							x	
6	*Paronychia camphorosmoides*			x						
7	*Paronychia chilensis* subsp. *chilensis* var. *chilensis*			x		x	x	x		
	*Paronychia chilensis* subsp. *chilensis* var. *mutica*					x				
	*Paronychia chilensis* subsp. *subandina*					x				
8	*Paronychia coquimbensis* subsp. *coquimbensis* var. *coquimbensis*					x				
	*Paronychia coquimbensis* subsp. *coquimbensis* var. *appressa*					x				
9	*Paronychia communis* subsp. *communis* var. *chicligastensis*	x								
	*Paronychia communis* subsp. *communis* var. *communis* f. *communis*		x	x			x	x	x	
	*Paronychia communis* subsp. *communis* var. *communis* f. *subglabra*				x					
	*Paronychia communis* subsp. *communis* var. *pungentifolia*	x		x						
10	*Paronychia compacta* subsp. *compacta*		x						x	
	*Paronychia compacta* subsp. *boliviana*		x							
	*Paronychia compacta* subsp. *purpurea*								x	
11	*Paronychia ellenbergii*								x	
12	*Paronychia franciscana*					x				
13	*Paronychia glabra*								x	
14	*Paronychia hartwegiana*						x		x	
15	*Paronychia hieronymi*	x	x							
16	*Paronychia johnstonii*					x				
17	*Paronychia jujuyensis*	x								
18	*Paronychia libertadiana*								x	
19	*Paronychia limaei*								x	
20	*Paronychia macbridei*								x	
21	*Paronychia mandoniana*		x						x	
22	*Paronychia membranacea*								x	
23	*Paronychia microphylla*	x	x			x				
24	*Paronychia muschleri*		x						x	
25	*Paronychia peruviana*		x							
26	*Paronychia revoluta*			x						
27	*Paronychia sanchez-vegae*								x	
28	*Paronychia setigera*	x	x	x						
	*Paronychia setigera* subsp. *cordobensis*	x								
	*Paronychia setigera* subsp. *setigera* var. *longiseta* subvar. *longiseta*	x								
	*Paronychia setigera* subsp. *setigera* var. *longiseta* subvar. *subglabra*	x								
29	*Paronychia ubinensis*		x						x	
30	*Paronychia weberbaueri*								x	
**Number of species per country**		2	3	17	10	7	9	12	2	1

## Data Availability

Not applicable.

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
