# Peer review of "The Genus Paronychia (Caryophyllaceae) in South America: Nomenclatural Review and Taxonomic Notes with the Description of a New Species from North Peru"

_plants, 2023, doi:10.3390/plants12051064_

Round 1

Reviewer 1 Report

The authors have done a great work about the revision of the typification status of Paronychia taxa in South America. The search of the potential vouchers as type material was very thorough, and the findings of new materia is quite relevant for typification purposes. The typification tasks and the proper use of the taxonomic rules based on the ICN were adequately implemented. The selection of the proposed vouchers as type material is perfectly adequate.    

However, some flaws are still found throughout the paper, and I would recommend to resolve them to get the paper finally published.

General comments

  • Since the ICN taxonomic rules has been properly used for the typification of the taxa, the taxonomic names of the many infraspecific taxa of Paronychia (i.e. subspecies, varieties and forms) are not properly written accordingly to the ICN rules. For example, as the authors know, the corresponding taxonomic name of the subspecies should be written as P. brasiliana subsp. brasiliana or P. brasiliana subsp. pubescens, and not simply as “brasiliana” or “pubescens” . This incorrect use of the denomination of subspecies, varieties or forms is widely use in the paper (e.g. subsp. chilensis instead of P. chilensis subsp. chilensis, or subsp. subandina instead of P. chilensis subsp. subandina). More cases are easily found for any infraespecific taxa, and I highly recommend to review it and all of them to write them following the ICN. The infrageneric taxa also should be written accordingly to ICN rules, so themes of the sections must be preceded by the name or abbreviation of the genus (i.e. page 2, P. sect. Paronychia, and not simply as sect. Paronychia). Moreover, the recommended writing to abbreviate the word form is ‘f.’ and not ‘fo.’ (see Art 5, ICN).       
  • Sometimes some misspellings are found within the manuscript, as ‘compacta’ in small letters (see Table 1), herbairum (p. 7), typificaiton, etc. Review the whole manuscript.
  • In some mentioned specimens, a special subsection is written as ‘Specimen examined’ and ‘Specimen evaluated’… why have you denoted this difference? This material is not used for typification purposes, and this material is not used for any specific purposes in the paper. Why do you have included it? There is not a descriptive section using morphological features based on these vouchers. The current paper is uniquely based on nomenclature and typification, and thus, the information of the studied material for typification should be included and mentioned with their corresponding info.       

Introduction

  • page 2. About the number of 28 species and 21 infrageneric taxa, this latter really means infrageneric or is it infraespecific? 
  • The main objectives of this paper are not clear enough since, the paper corresponds entirely to the typification of the names, and there is not a morphological comparative study of the species, and specially about those species closed related to the new proposed species, and there is not information about the morphological features os each species. It is evident that the authors have more information and data that those finally included in this current version, and sometimes the authors state some sentences not properly supported. Therefore, I would recommend to change or exposed adequately the objetives of this paper to be adjusted to the content of the paper.

Material and methods

  • If the main objetive of this paper is to give and deeply revision of the typification of the names of Paronychia in South America, the indication of additional material of each species is not really necessary, since this material is not used for typification purposes.  In addition, the indication of field surveys are not really needed for typification purposes. 
  • The proposal of the new species is not explained in this section, so which species were used to be compared with, the studied material, and those morphological features analyzed to support the final description. Are the morphological features used for any specific part of the typification study?          

Results and discussion

  • page 2. In the first paragraph, there is a mention that 30 species are recognized, and in this parer a new species is described, and later, surprisingly   there is a mention that only 29 species are finally recognized. Without previous explanations, this parte of the paragraph is quite confuse. Review and clarity better what you want to expose. 
  • Table 1. The information of the columns can not be read adequately. I would recommend to use the official abbreviations of each country, and to write the meaning of each one within the table caption. In addition, the included species in this table 1, are they the final results of this paper or do they correspond to the species mentioned in the literature? It is not clear. 
  • Figure 1: after writing the whole named of the genus at the beginning of the figure caption, it is not necessary to use it again for the following names of the species in the same figure caption.   
  • P. arbuscula: the link of the proposed voucher from P as material type seems to be incorrect, since it mentions another voucher.  
  • P. arequipensis: p.4 - write the name of the species and subspecies in italics. line10 - ‘arequipensis’ not properly written based on ICN. The both writings are found ‘arequepensis’ and ‘arequipensis’, write this words in italics and between ‘’, so the words can be easily distinguished from the main text. Why is the species prone to decline? Give some explanation.  
  • P. brasiliana: P. bonariensis in italics; nom. superfl. et illeg. also in italics; write in italics and bold all the subspecies (including the typical one); review the writing of the subspecies (see ICN, and general comments); does the difference between the two subspecies based only on the distribution areas? are there more morphological information? 
  • P. comphorosmoides: the voucher P ends in 23, what is it? an isolectotype? Clarify it.   
  • P. chilensis: write the name of the subspecies, varieties names accordingly to ICN (see general comments), and review the whole paragraph; some words are misspelling as herbarium; the mention of the subspecies and varieties (see 7a, 7b, 7a1, etc.) are not written in italics and bold. 
  • P. coquimbensis: the word lectotype in capital letters (similarly as have been done for other species); include the mention and information of those vouchers potentially related to be elegible as lectotypes; 8a - the word ‘var.’ not in italics and bold. 
  • P. communis: starting the sentences with ‘[11]’ seems quite strange, and I would suggest to add the surnames of the authors and then, add [11]; the three varieties of P. communis are not adequately written based on ICN. What are the authors of the P. communis var. chicligastensis? The name of the varieties are written sometimes in bold and others not (see 9a); the ‘f.’ must not be written in italics. 
  • P. glabra: the word ‘from’ should not be written in italics.    
  • P. hartwegiana: within the specimens evaluated, the authors have included the information of that voucher proposed as lectotype, but not the information about the remaining vouchers.
  • P. hieronymi: add the link to see the voucher (as you have been done for the previously names). 
  • P. jujuyensis: the authors mention type material, but which type?
  • P. johnstonii: the var. scabrida not properly written (see general comments); the authors state some synonyms but not references or morphological studies are given ti support it. is it only based on bibliography? Give some explanation.
  • P. mandoniana: a comma is missed between the author of the species and the original journal (Linnaea), and it is missed in some other cases, so review the whole paper.  
  • P. menbranacea: in the synonyms, there is a sentence that it is not understandable. Please, review it.    
  • P. muschleri: is something missing between ‘pro and Paronychia’?
  • P. compacta: a full-stop is missed between pro and Paronychia, and thus, the word Paronychia is bad written (as Paronichia); nom. illeg. in italics; I would recommend to change the format to add and expose the relevant specimens since to is quite strange at the current form. 
  • P. sanchez-vegae: there are not codes for the vouchers of certain herbaria, as B, but previously the authors include the specific code for vouchers of this herbarium. Review the whole paper.      
  • P. setigera: the selected lectotype does not show a link to see the material, but this voucher belong to K, and it seems strange not include the link. Review the mentions of subspecies and varieties according to ICN; do not add the name of the species for the naming of the subsection of specimens evaluated; 28a, the names of the taxon is not in italics neither in bold.
  • Species excluded from South America: the info about the three vouchers from MO are not given and more data should be added.    

Author Response

Dear Editor, reviewers, 

Attached the response letter. 

Best regards, 

Reviewer 2 Report

The manuscript covers an interesting topic concerning a nomenclatural and taxonomic study of south American Paronychia.

My major concerns are about the putative new species such as P. glabra and P. jujuyensis. In the first case, at least some appropriate statistical analysis concerning morphological data, in comparison with P. hieronymi and P. macbridei are needed. In the second case, "We here propose to treat it as separate species" is definitely a too scarce justification to erect a new combination at species level.

More precise, quantitative and convincing evidence should be provided in these cases.

A similar problem, albeit there at least some justification is provided, concerns P. arequipensis.

I also annotated directly the manuscript highlighting several typos and minor comments/corrections.

Author Response

Dear Reviewer,

Responses in the attached document. 

Round 2

Reviewer 1 Report

Thanks to the authors, since the manuscript has improved and the authors has fully resolved most of the proposed comments. However, there are some minor comment/corrections that should be resolved. 

  • In the introduction, add the ‘P.’ (of Paronychia) before the mention of the tribe and section (i.e. P. sect. Paronychia. P. tribe Paronychieae), so the mention of the infrageneric taxa follows ICN.
  • In material and methods, and about the description of new species, I would recommend to give more information about the comparison study, that you have done, especially about which species is related to (or add any information in introduction, e.g. for what species it could be confused to?). The taxonomic discussion is mostly based on P. hieronymi. 
  • Although some MO vouchers do not have code, the specific information of the vouchers should be added (i.e. specimens examined, similarly as you have done for all the Paronychia species), so any further researcher might know about which sheet you have done your statements about the non-presence of Paronychia species from South America.   
  • P. setigera: check the specimens examined and delete the mention of ‘for Paronychia setigera’ (since this mention is only done for this species, and none for any other additional species). 
  • The abbreviation of versus (‘vs.’) appears with italics and without italics throughout the manuscript. Review the whole paper. 
  • The writing of the months of the examined specimens appears with two versions: in capital letters and small letters. Review the whole paper. 
  • P. glabra: Review the material included as paratypes. You have mentioned some specimens as holotypes!!!
  • P. franciscana: is the link of the isolectotype (LS589370) correct?  The link mentions other voucher. 
  • P. franciscana: What is the difference between LP036403(!) and LP036403! (related to the situation of the exclamation mark)?
  • P. chilensis: Review the second paragraph, since a mention of P. chilensis should be deleted.  
  • P. arequipensis: Review the paragraph Notes. In the sentence “P. arequepensis and Paronychia microphylla differ each other by the habit (chamaephyte up to 30 cm ….“ the name of the genus is abbreviated for the first species and not for the second one. In addition, and about the compared morphological data between P. arequipensis  and P. microphylla, I would recommend to follow the same order of mention of the species. For the habit: it is P. microphylla vs. P. arequipensis; in shape of leaves: it is P. arequepensis vs. P. microphylla; for the length of the sepals: P. microphylla vs. P. arequipensis; and for size of leaves,  is it P. microphylla vs. P. arequipensis?   
  • Table 1. The word form is abbreviated sometimes as ‘fo.’ Change it as ‘f.’
  • Finally, I would recommend to the authors to follow ICN for the name of taxa below the rank of species: See art. 24.1 The name of an infraspecific taxon is a combination of the name of a species and an infraspecific epithet. A connecting term is used to denote the rank.” The authors have properly followed the ICN code for the selection of types, and hence, I would still recommend to follow the ICN, about writing properly the name of the varieties throughout the ms. The ICN should be followed for all nomenclatural and taxonomic aspects.   

Author Response

Dear Reviewer,

Attached are the responses to each marked point. 

Best regards, 

The authors

Reviewer 2 Report

My comments and doubts dealing with Paronychia glabra were not addressed at all, and only partially addressed concerning P. jujuyensis.

Minor problems:

"reaches 30", not "reaches to 30"

the sections "results and discussion" should be renamed as I already suggested in my previous review

All the part dealing with P. compacta still shows problems:

-"should be corrected", not "would be corrected"

-months which should be written with starting capital

Author Response

Dear Editor/Reviewer, 

Attached is the version corrected (with corrections highlighted) and replies to the reviewers. 

Daniel 

Round 3

Reviewer 2 Report

My comments and doubts dealing with Paronychia glabra (expressed already in the first review round) were not yet addressed at all, and only partially addressed concerning P. jujuyensis.

Author Response

Dear reviewer, 

Attached our corrections and response. 

Best regards, 
